# Concentration, temporal variation and sources of black carbon in the Mount Everest region retrieved by real-time observation and simulation

Xintong Chen [1, 4], Shichang Kang [1, 2, 4], Zhiyuan Cong[2, 3], Junhua Yang [1], Yaoming Ma [3]

[1]State Key Laboratory of Cryospheric Science, Northwest Institute of Eco-Environment and Resources, Chinese Academy of Sciences, Lanzhou 730000, China

[2]CAS Center for Excellence in Tibetan Plateau Earth Sciences, Chinese Academy of Sciences, Beijing 100101, China

[3]Key Laboratory of Tibetan Environment Changes and Land Surface Processes, Institute of Tibetan Plateau Research, Chinese Academy of Sciences, Beijing 100101, China

[4]University of Chinese Academy of Sciences, Beijing 100049, China

*Correspondence to*: Shichang Kang (shichang.kang@lzb.ac.cn)

**Abstract.** Based on the high-resolution measurement of black carbon (BC) at Qomolangma (Mt. Everest) Station (QOMS, 28.36°N, 86.95°E, 4276 m a.s.l.) from 15 May 2015 to 31 May 2017, we investigated the seasonal and diurnal variations in BC and its potential source regions. Both monthly and daily mean BC concentrations reached the highest values in the pre-monsoon season and the lowest values in the monsoon season. The highest monthly and daily mean BC concentrations were at least one order of magnitude higher than the lowest concentrations. For the diurnal variation, the BC concentrations remained significantly high from late night to morning in the pre-monsoon season. Meanwhile, the westerly winds prevailed during this period, implying the potential for pollutants to be transported across the Himalayas from long-distance sources to QOMS along the valley. In the monsoon season, the BC concentrations remained low but peaked in the morning and in the noon, which might be caused by local emissions from cooking. By analyzing the simulation results from the backward trajectories of air masses and the fire spot distribution from the MODIS data, we found that the seasonal cycle of BC was significantly influenced by the atmospheric circulation and combustion intensity in the Mt. Everest region. The transport mechanisms of BC were further revealed using a WRF-Chem simulation during severe pollution episodes. For the pollution event in the monsoon season, BC aerosols in South Asia were uplifted and transported to the Mt. Everest region by the southerly winds in the upper atmosphere. However, for the events in the pre-monsoon season, BC from northern India was transported and concentrated on the southern slope of the Himalayas by the northwesterly winds in the lower atmosphere and then transported across the Himalayas by the mountain-valley wind. Relatively less BC from northwestern India and Central Asia was transported to the Mt. Everest region by the westerly winds in the upper atmosphere.

## 1 Introduction

Black Carbon (BC), mainly from the incomplete combustion of fossil fuels or biomass, has drawn much attention due to its influences on the environment and human health (Bond, 2004; Ramanathan et al., 2005; Anenberg et al., 2012) and is seen as an important factor that may lead to global warming, in addition to greenhouse gases (Hansen et al., 2000; Jacobson, 2002;

Bond et al., 2013; Ramanathan and Carmichael, 2008). BC can substantially absorb solar radiation and causes atmospheric
heating (Jacobson, 2001; Ramanathan et al., 2005; Ji et al., 2015). Moreover, BC can be suspended as fine particles in the
atmosphere for approximately one week, be transported far away from its emission sources, and then be removed by dry and
wet deposition (Oshima et al., 2012; Cooke et al., 2002; Jurado et al., 2008). When BC is deposited on snow and ice, it can
significantly reduce surface albedo (Flanner et al., 2007; He et al., 2017) and accelerate glacier and snow cover melting,
causing an impact on the regional climate, hydrology, and water resources (Li et al., 2018; Ming et al., 2008; Ramanathan and
Carmichael, 2008).

The Tibetan Plateau (TP), generally known as the "Third Pole", is the highest plateau with a large number of glaciers and

snow cover (Kang et al., 2010; Lu et al., 2010; Yao et al., 2012). Even though the TP is a remote region with few affects from
anthropogenic activities, previous observations have indicated that BC is an important contributor to the rapid shrinking of
glaciers over the TP via decreasing surface albedo and atmospheric warming (Xu et al., 2009; Yang et al., 2015; Li et al., 2017b;
Zhang et al., 2017b; Qu et al., 2014; Ji, 2016; Xu et al., 2016; Lee et al., 2017). Moreover, previous studies have also suggested
that the emissions from South Asia and East Asia are the major sources of BC on the TP (Li et al., 2016a; Lu et al., 2012; He
et al., 2014b; Zhang et al., 2015; Yang et al., 2018), and the high emissions from South Asia can be transported across the
Himalayas and further to the inland TP (Luthi et al., 2015; Xu et al., 2014; Cong et al., 2015a; Kang et al., 2016; Wan et al.,
2015). Meanwhile, the seasonality of BC aerosols is closely related to atmospheric circulation that helps to bring the BC
aerosols across the Himalayas (Cong et al., 2015a; Cong et al., 2015b; Yang et al., 2018). Additionally, a large number of
studies have demonstrated that the BC and dust from Central Asia and northern Africa could also be transported to the TP
(Wang et al., 2016; Lu et al., 2012; Zhao et al., 2012; Wu et al., 2010; Zhang et al., 2015).

51       Mt. Everest could be regarded as a very sensitive area under the influence of BC aerosols. Previous research on

atmospheric BC in the Mt. Everest region was mainly based on the thermal/optical analytical method, using quartz filter
samples (Cong et al., 2015a). However, there is still a lack of investigations on the diurnal and seasonal variations in BC in
this region. Therefore, to fill such gaps and understand the variations in and sources of BC in the pristine region, there is a
need for an efficient approach and additional studies. The Aethalometer can provide real-time high-resolution observation data
on the BC concentration, which is very important and necessary to better depict the characteristics of BC and its effects on the
environmental change.

In comparison with the observations, numerical models can better represent the atmospheric physical and chemical

processes. Many studies have used global climate models (GCMs) and chemical transport models (CTMs) to investigate the
origin and transportation of BC over the TP (Lu et al., 2012; Zhang et al., 2015; Menon et al., 2010; Kopacz et al., 2011; He
et al., 2014a). However, due to the coarse resolution, it is difficult for the CTMs and GCMs to capture the surface details of
the TP (Ji et al., 2015; Gao et al., 2008). Regional climate models (RCMs) can compensate for the shortcomings of coarser
global model grids by high-resolution simulations. In recent decades, RCMs have been developed to include multiple modules
and atmospheric chemistry processes. In addition, the advanced regional climate-chemistry model, Weather Research and
Forecasting (WRF) model (Skamarock et al., 2005) coupled with chemistry (WRF-Chem) has been successfully applied for
air quality research on the TP (Yang et al., 2017; Yang et al., 2018).
Here, we present real-time data of the BC concentration measured by the new Aethalometer model AE-33 from 15 May
2015 to 31 May 2017. The observed results are used to characterize the temporal variation and provide important information
on the possible sources and transport mechanisms of BC. By combining high-resolution measurements of the BC concentration
and the WRF-Chem model, we investigated the concentration level, temporal variation, and sources of BC in the Mt. Everest
region. The purpose of this study is to understand the impact of trans-boundary atmospheric BC on the Mt. Everest region and
depict the transport pathways of BC at different spatiotemporal scales.
**2 Materials and methods**
**2.1 Sampling site and meteorological conditions**
Mt. Everest (27.98°N, 86.92°E, 8844 m a.s.l.), the summit of the world, is located in the central Himalayas. The southern
slope of Mt. Everest is adjacent to the Indian continent, and the climate is warm and humid under the influence of the Indian
summer monsoon. Conversely, the northern side is cold and dry since the warm and humid airflow cannot reach it.
Qomolangma (Mt. Everest) Station for Atmospheric and Environmental Observation and Research, Chinese Academy of
Sciences (QOMS, 28.36°N, 86.95°E, 4276 m a.s.l.) (Fig. 1) is located on the northern slope of Mt. Everest, which was
established for continuous monitoring of the atmospheric environment (Cong et al., 2015a; Ma et al., 2011).
The meteorological parameters, i.e., air temperature, air pressure, humidity, wind speeds and wind direction, were
recorded by an automatic weather station at QOMS with 10 min time intervals. Meanwhile, the precipitation data were
collected by artificial measurement, as shown in Fig. S1. The entire year was divided into four seasons according to the Indian
monsoon transition characteristics, which includes pre-monsoon (March to May), monsoon (June to September), post-
monsoon (October to November), and winter (December to February) (Praveen et al., 2012; Zhang et al., 2017a). A clear
seasonal cycle of temperature and humidity can be observed in Fig. S1. Specifically, the temperature was high during the
monsoon season and low during winter, with a maximum in July and a minimum in January. Humidity followed a similar trend,
with high values from late July to early August and low values from December to February. During the observation period, the
wind speed increased significantly from November to April. The wind direction at QOMS is affected by the local topography,
which consists of a series of small valleys. During the pre-monsoon season (dry period), the westerly and southerly winds
begin to develop and play an important role in atmospheric pollution circulation. However, during the monsoon season, the
southwesterly winds prevail and bring much moisture from the Indian Ocean to the Mt. Everest region, increasing the humidity
and precipitation. With the retreat of the monsoon, the southwesterly winds decrease and the prevailing wind direction changes
to westerly and northeasterly in winter with limited moisture (Fig. S1).

## 2.2 BC measurements

There are several available methods capable of measuring BC concentrations, and these methods can be classified into
three categories. First is the thermal/optical method, which uses a quartz filter to collect aerosols, and they are thermally
volatilized in several temperature steps (Schauer et al., 2003). The signals of evolving carbon measured by thermal/optical
transmission (TOT) or thermal/optical reflectance (TOR) can be converted to the concentration of BC (Chow et al., 1993;
Chow et al., 2001). However, the time difference between sampling and detection, the impact of mineral dust, and the
determination of the split between organic carbon (OC) and elemental carbon (EC, the same as BC) can cause deviations (Li
et al., 2017a; Schauer et al., 2003). The second category is the technique of the single particle soot photometer (SP2), which
can quantify BC by laser-induced incandescence because BC is the predominant refractory absorbing aerosol, which can be
heated by an intense laser beam and emit significant thermal radiation (Stephens et al., 2003). This method measures the mass
of BC in individual particles, but the accuracy depends on the selected calibration material (Schwarz et al., 2010; Laborde et
al., 2012). Finally, the optical method measures the reduction in light intensity induced by BC aerosols collected on the
sampling medium (Hansen et al., 1984; Petzold and Schonlinner, 2004). The Aethalometer is a widely used instrument based
on the optical method that can provide real-time BC concentration measurements, but all filter-based optical methods exhibit
loading effects that can lead to the underestimation of BC concentrations (Bond et al., 1999; Virkkula et al., 2007; Park et al.,
2010; Hyvarinen et al., 2013; Drinovec et al., 2015). However, the newly developed Aethalometer model AE-33 uses a real-
time loading effect compensation algorithm that can provide high-quality data, which is very helpful for the accurate
determination of BC concentrations and source apportionment (Drinovec et al., 2015).
Therefore, the airborne BC concentrations at QOMS were monitored by the new Aethalometer model AE-33 (Magee
Scientific Corporation, USA). The instrument was set in an indoor room with an inlet installed at approximately 3 m above
the ground level and was operated at an airflow rate of 4 LPM with a 1 min time resolution. AE-33 has seven fixed wavelengths
(i.e., 370, 470, 520, 590, 660, 880 and 950 nm), which can acquire the BC concentration according to the light absorption and
attenuation characteristics from the different wavelengths (Hansen et al., 1984; Drinovec et al., 2015). Generally, the BC
concentration measured at 880 nm is used as the actual BC concentration in the atmosphere, as the absorption of other species
of aerosols is greatly reduced in this wavelength (Sandradewi et al., 2008a; Sandradewi et al., 2008b; Fialho et al., 2005; Yang
et al., 2009; Drinovec et al., 2015). Compared to previous Aethalometer models, AE-33 uses dual-spot measurement and a
real-time calculation of the "loading compensation parameter", which can compensate for the "spot loading effect" and obtain
high-quality BC concentration (Drinovec et al., 2015). The main structure of this algorithm is as follows:
BC (reported)=BC(zero loading)×$(1 - k\mathrm{ATN})$                    (1)
$\mathrm{ATN} = -100 \ln(I/I_0)$                    (2)
BC1=BC×(1 − $k$ATN1)                                                    (3)
BC2=BC×(1 − $k$ATN2)                                                    (4)
where BC (reported) is the uncompensated BC concentration; BC (zero loading) is the desired ambient BC value that would
be obtained in the absence of any loading effect; $k$ is the loading effect compensation parameter; $I$ and $I_0$ are the light intensity
of the measurement spot and reference spot; and ATN is the attenuation of light through filter tape. The BC component of the
aerosols is analyzed on two parallel spots drawn from the same input stream in AE-33 but collected at different rates of
accumulation. This means that we can obtain different ATN but the same loading parameter $k$ (Drinovec et al., 2015).
Combining Eq. (3) and Eq. (4), the compensation parameter $k$ and the desired value of BC compensated back to zero loading
can be calculated. Based on the dual-spot technology, the new real-time compensation algorithm allows extrapolation to zero
loading and obtains the accurate BC concentration (Drinovec et al., 2015; Crenn et al., 2015; Zhu et al., 2017). Previous studies
have evaluated the real-time compensation algorithm of dual-spot Aethalometer model AE-33 and indicated that AE-33 agrees
well with the post-processed loading effect compensated data obtained using earlier Aethalometer models and other filter-
based absorption photometers, implying the good performance of this new algorithm (Drinovec et al., 2015; Rajesh and
Ramachandran, 2018).
**2.3 Model simulation and datasets**

WRF-Chem version 3.6 was used to analyze the spatial distribution, transport mechanism, and source apportionment of

BC during the four observed pollution episodes. The WRF-Chem model is an expansion of the WRF meteorological model
and considers complex physical and chemical processes such as emission and deposition, advection and diffusion, gaseous and
aqueous chemical transformation, and aerosol chemistry and dynamics (Grell et al., 2005). Here, the numerical experiments
were performed at a 25 km horizontal resolution with 122 and 101 grid cells in the west-east and north-south directions,
respectively. The simulated domain was centered at 25°N, 82.5°E and had a 30-layer structure with the top pressure of 50 hPa.
The key physical and chemical parameterization options for the WRF-Chem model were based on a previous study on the TP
(Yang et al., 2018). The initial meteorological fields were taken from the National Centers for Environmental Prediction (NECP)
reanalysis data with a horizontal resolution of 1° × 1° at 6 h time intervals. The anthropogenic emission inventory was obtained
from the Intercontinental Chemical Transport Experiment-Phase B (INTEX-B) (Zhang et al., 2009) with a resolution of 0.5°
× 0.5°. The biogenic emissions were obtained from the Model of Emission of Gases and Aerosol from Nature (MEGAN)
(Guenther et al., 2006), and the fire emissions inventory was based on the fire inventory from NCAR (FINN) (Wiedinmyer et
al., 2011). Additionally, the Model for Ozone and Related chemical Tracers (MOZART, http://www.acom.ucar.edu/wrf-
chem/mozart.shtml) (Emmons et al., 2010) dataset was used to create improved initial and boundary conditions for the BC
simulations during these pollution episodes.

Furthermore, to predict the source region of BC, we used the Hybrid Single-Particle Lagrangian Integrated Trajectory

(HYSPLIT-4) model to calculate the backward trajectories of the air masses (Stein et al., 2015), and the calculation data was
obtained from the National Centers for Environmental Prediction/National Center for Atmospheric Research (NCEP/NCAR)
reanalysis data ($2.5° \times 2.5°$). The parameter settings for the backward trajectory calculation in the HYSPLIT model were
chosen according to a previous study in this area (Xu et al., 2014). The active fire product provided by the Fire Information
for Resource Management System (FIRMS, https://firms.modaps.eosdis.nasa.gov/firemap/) was chosen to investigate the
biomass burning emissions over the region in different seasons.

## 3. Results and discussion

### 3.1 Temporal variations in BC

#### 3.1.1 Monthly variation in BC

The monthly mean BC concentrations at QOMS are shown in Fig. 2a. There was a significant increase in the BC
concentrations in winter, and the highest value occurred during the pre-monsoon season ($923.1 \pm 685.8$ ng/m$^3$ in April).
Meanwhile, during the monsoon, lower BC concentrations were recorded, and the lowest value was observed in July ($88.5 \pm$
$29.8$ ng/m$^3$). This seasonal change was consistent with the previous studies of elemental carbon (EC or BC) at Nepal Climate
Observatory-Pyramid station (NCO-P, 27.95°N, 86.82°E, 5079 m a.s.l.) from March 2006 to February 2008 (Fig. 2b) (Marinoni
et al., 2010) and at QOMS from August 2009 to July 2010 (Fig. 2c) (Cong et al., 2015a), indicating a similar BC source
between the southern and northern sides of the Himalayas. As EC was sampled by quartz filters and detected using the
thermal/optical analytical method in previous studies, there may be some differences in the values of EC compared to those of
BC, for instance, the overestimation of EC due to the potential effect of carbonates in mineral dust of the samples when using
the thermal/optical method (Li et al., 2017a). The monthly variation in EC at Nam Co Monitoring and Research Station for
Multisphere Interactions (Nam Co station, 30.77°N, 90.98°E, 4730 m a.s.l.) from January to December during 2012 (Fig. 2d)
(Wan et al., 2015) also showed a similar variation, but the peak value of EC occurred in winter. Additionally, the monthly mean
EC concentrations at Nam Co station were generally lower than those at QOMS, suggesting that the impact of the
anthropogenic activities on the inland TP was weaker than that on the south edge of the TP. Previous studies have demonstrated
that the influence of polluted air masses from the "Atmospheric Brown Clouds" over South Asia could reach the southern
foothills of the Himalayas and that the mountain-valley breeze circulation carried the polluted air masses onto the TP (Luthi
et al., 2015; Cong et al., 2015a; Bonasoni et al., 2008; Yang et al., 2018). Therefore, the seasonal cycle of BC concentrations
at QOMS was likely affected by the atmospheric circulation and the emissions from South Asia, which will be further explained
in Section 3.3.

#### 3.1.2 Daily variation in BC

Fig. 3 shows the daily mean BC concentrations at QOMS, which present a significant seasonal pattern, with a maximum during the pre-monsoon season (2772.3 ng/m$^3$) and a minimum during the monsoon season (36.4 ng/m$^3$). During the monsoon season, the BC concentration was observed to be lower than 150 ng/m$^3$, but it gradually increased during the post-monsoon and winter. The mean concentration of daily BC at QOMS was 298.8 ± 341.3 ng/m$^3$, which was close to the previous result (250 ± 220 ng/m$^3$) (Cong et al., 2015a).

The comparison between daily mean BC concentrations (Fig. 3) and the meteorological parameters (Fig. S1) suggested that the increasing precipitation during the monsoon led to the washout of atmospheric particles, promoting the wet deposition of BC. This process caused a decrease in BC concentrations during the monsoon, representing the background level during the period. The prevailing wind direction was southwesterly during the monsoon period and westerly during the non-monsoon periods. Therefore, the variations in BC might be linked to the influence of meteorological conditions and the contribution of long-distance transport from urbanized areas to QOMS. Moreover, it cannot be ignored that there were continuous high concentrations of BC above 1000 ng/m$^3$ during 8-10 June 2015, 19-22 March 2016, 9-30 April 2016, and 11-14 April 2017, indicating that the heavy pollution episodes happened at QOMS during those days. A detailed analysis of these pollution events is presented in Section 3.4.

**3.1.3 Diurnal variation in BC**

Diurnal variation characteristics can be used to analyze the impact of local meteorological processes and anthropogenic activities on the BC concentrations at QOMS. The half-hourly mean BC concentrations are presented in Fig. 4. In the pre-monsoon season, the diurnal BC concentrations remained significantly high from late night to morning (00:00-12:00 BJT (Beijing Time: UTC+8h), which is two hours earlier than local time) and increased gradually after the lowest value at approximately 15:30 BJT. Elevated BC concentrations were also observed in the afternoon during the post-monsoon and winter periods, and high BC concentrations occurred from late night to morning. The BC concentrations during the monsoon season were significantly lower than those during the other seasons but peaked in the morning (08:00 BJT) and in the noon (14:00 BJT). Previous studies have demonstrated that the local wind system on the northern slope of Mt. Everest is composed of a morning "valley wind", a late morning-afternoon "glacier wind" weakened by "valley wind", and an evening-early night "mountain wind"(Zou et al., 2008). The QOMS is located in the s-shape valley north of Mt. Everest (Ma et al., 2011). The glacier wind and down mountain wind from the south developed in the afternoon and at night, which provided the potential possibility for pollutants from long-distance sources transported to QOMS along the valley and increased the BC concentrations in the non-monsoon periods. The valley wind from the north in the morning could bring the short-distance emissions from cooking or heating in several villages that are located north (approximately 5 km away) of QOMS. The BC concentrations were remarkably low in the monsoon season but peaked in the morning and in the noon, which might be due to the local emissions carried by the valley wind from the north.

To explain the significant high values from late night to morning (00:00-12:00 BJT) in the pre-monsoon season, the wind direction frequency at QOMS during 00:00-12:00 BJT and 12:00-24:00 BJT are presented in Fig. 5. During the sampling period in the pre-monsoon season, W (west) winds prevailed from late night to morning (Fig. 5a), accounting for 18.1% of the total wind directions, followed by ENE (east-northeast) winds (16.4%). This is consistent with the discussion above that there are potential impacts on the BC concentrations at QOMS from long-distance human activity emissions, which can be carried by the westerly winds, i.e., down mountain winds (Cong et al., 2015b). Moreover, the WRF-Chem simulation results showed that the profile of equivalent potential temperature (EPT) increased with altitude and the planetary boundary layer height (PBLH) and wind speed were much lower from late night to morning (Fig. S2), indicating a more stable atmosphere that obstructs the diffusion of BC aerosols. ESE (east-southeast) and NE (northeast) winds prevailed from late morning to night (12:00-24:00 BJT) (Fig. 5b), accounting for 17.6% and 15.3% of the total wind directions, respectively, implying a strengthened glacier wind or mountain wind (from the south), which caused the increase in BC contributed by long-distance sources. During the pre-monsoon season, the strong mountain winds and glacier winds could transport large amounts of trans-Himalayan pollution from heavily polluted areas of South Asia to QOMS; therefore, the long-distance sources play a major role in the diurnal variation in the BC concentrations at QOMS during this period.

**3.2 Comparison of the BC concentrations with other sites on the TP**

A previous study has revealed that low BC concentrations in China can be found on the TP, with values of approximately 200-1000 $ng/m^3$ in $PM_{2.1}$ and 300-1500 $ng/m^3$ in $PM_{9.0}$ (Xin et al., 2015). To better understand the BC loading level, we compared our results with previous studies from other locations over the TP. As shown in Fig. 1, the BC concentrations at Muztagh Ata and Qilian Shan presented low values, which can be regarded as the background concentration level for inland Asia (Cao et al., 2009; Zhao et al., 2012). In contrast, the BC concentration at Lhasa city was higher than that at other sites on the TP, which was mainly contributed by local fossil fuel combustion (Li et al., 2016b). In addition, under the impact of the long-range transport of anthropogenic emissions from the east and significant dust input from the west, the BC concentration at Qinghai Lake also showed a relatively high value (Li et al., 2013). The BC concentration at Beiluhe was slightly higher than that at Qinghai Lake, mainly from the arid regions in northwestern China in spring and from the southern slope of the Himalayas in winter (Wang et al., 2016). Therefore, the long-range transportation from Central Asia and East Asia contributed greatly to the BC aerosols in the northern TP. For the sites in the central and southeastern regions on the TP (e.g., Nam Co and Ranwu), which are isolated from anthropogenic activities with relatively clean atmospheric environments, the BC concentrations at these two sites were above 130 $ng/m^3$, likely due to the influence of long-range transport from South Asia (Wan et al., 2015; Wang et al., 2016). Compared with the locations on the southern slope of the Himalayas (e.g., NCO-P and Manora Peak), the BC concentration at QOMS was close to that at NCO-P but much lower than that at Manora Peak, which is near the polluted areas in South Asia and largely affected by anthropogenic emissions (Marinoni et al., 2010; Ram et al., 2010).

This implies that the combustion emissions from South Asia affect not only the lower latitudes in the vicinity but also the
higher latitudes in the Himalayas and the interior of the TP due to long-range transport.

**3.3 Potential sources and transport mechanisms of BC in different seasons**

The seasonal variation in the BC concentrations was correlated with the combustion intensity of sources and atmospheric
circulation. The "Atmospheric Brown Clouds" over South Asia contain large amounts of aerosol components such as the high
loading emissions of BC from biomass burning, which can reach the TP within a few days (Ramanathan et al., 2005;
Ramanathan and Ramana, 2005; Luthi et al., 2015). A previous study has quantified biomass burning sources contributing to
BC aerosols from the Himalayan region of Nepal and India and showed that the major fires were concentrated from March to
June; additionally, most fires occurred in the low elevation areas dominated by forests and croplands (Vadrevu et al., 2012).
Therefore, we further checked the biomass burning emissions in the Mt. Everest region and its vicinities using the active fire
product from the MODIS data during four seasons (August 2015 to April 2016) provided by the FIRMS (Fig. 6). It is clearly
shown that there were large numbers of active fire spots in northern and central India, Pakistan and Nepal in winter and in the
pre-monsoon season. Moreover, referring to Cong et al. (2015a), the active fire spots represent agricultural combustion and
forest fires in this region, which might substantially contribute to BC aerosols. During the monsoon season, insignificant fire
spots appeared in South Asia, representing less biomass burning in that period.
To further explore the sources and the long-range transport mechanism of BC aerosols at QOMS, we calculated the
frequency plots for 5-day backward trajectories arriving 1 km above ground level (Fig. 7). During the non-monsoon seasons,
the air masses were affected by the westerly winds. The air masses reaching the Mt. Everest region were mostly from the
northwest, indicating that the biomass burning emissions in Pakistan, northern India and Nepal could be transported to the Mt.
Everest region. However, for the difference in the combustion intensity, high concentrations of BC were found only during the
pre-monsoon season. During the monsoon season, the southerly winds dominated in the Mt. Everest region, and the air masses
were mainly from the Arabian Sea and the Bay of Bengal with substantial moisture. At this period, the precipitation on the
southern side of the Himalayas was above 1200 mm (Xu et al., 2014), which can improve the wet removal efficiency of BC.
Moreover, the biomass combustion emissions in South Asia in this period were very low. Therefore, the BC concentrations at
QOMS were close to the background level during the monsoon season. Meanwhile, the local meteorological conditions also
play a very important role in the transport of pollutants across the Himalayas from South Asia. Previous studies have shown
that the local wind system was mainly composed of up-valley wind on the southern slope and down-valley wind on the northern
slope, which facilitates the exchange of air between the bottom and upside of the atmosphere, and facilitates the coupling of
air flow between the southern and northern slopes, which allows the pollutants from South Asia to easily cross the Himalayas
and be transported to the TP from the valley (Zou et al., 2008; Chen et al., 2012; Cong et al., 2015b; Tripathee et al., 2017;
Dhungel et al., 2018).

### 3.4 Pollution episodes analysis by WRF-Chem modeling

In this section, we analyzed four pollution events with BC concentrations above 1000 ng/m³ in detail, including event A during 8-10 June 2015, event B during 19-22 March 2016, event C during 9-30 April 2016, and event D during 11-14 April 2017. Fig. 8 shows the spatial characteristics of the WRF-Chem modeled surface BC concentrations during the four pollution episodes. It can be seen that the high values of surface BC concentrations always appeared in South Asia, although the high-value centers changed in different pollution events. For event A, the most serious pollution appeared in Nepal and northern India. There was relatively less BC near Mt. Everest in event B than in the other events. However, for event C, the high BC concentration areas were mainly along the southern slope of the Himalayas in Nepal and eastern India, which can highly impact the BC concentrations in the Mt. Everest region. In event D, the high BC concentrations occurred in Nepal and some parts of India. To evaluate the model performance, the temporal variation in measured and simulated BC concentrations at QOMS during these four pollution episodes are displayed in Fig. S3. As shown in Fig. S3, for the four pollution episodes, the WRF-Chem model captured the variation trends of the observed BC concentrations, with correlation coefficients all above 0.8. This implies that the model could reproduce the distribution of BC concentrations in this region. Additionally, comparisons between the modeled wind and precipitation and the wind and precipitation from reanalysis data and in-situ observations indicated that the WRF-Chem model could capture the spatiotemporal variations in the meteorological elements (Fig. S4 and Fig. S5).

The sources and transport mechanisms of BC aerosols during these pollution episodes can be indicated by analyzing the air flow. Fig. 9 shows the variation in the BC concentrations and wind fields at different altitudes in the atmosphere (850 hPa, 500 hPa, and 200 hPa). For event A during the monsoon season, there was a cyclone in northern India at 850 hPa that moved near-surface BC aerosols upward, and then, the southerly winds at 500 hPa and 200 hPa transported the BC aerosols to the Mt. Everest region. For events B-D in the pre-monsoon season, the northwesterly winds prevailed in South Asia at 850 hPa and brought BC from northern India to the southern slope of the Himalayas, and the westerly winds at 500 hPa and 200 hPa transported relatively less BC from northwestern India and Central Asia to the Mt. Everest region. Previous studies also pointed out that BC can be transported across the Himalayas to the Mt. Everest region by the mountain-valley wind system (Zou et al., 2008; Cong et al., 2015b; Dhungel et al., 2018). Thus, we needed to further analyze the impact of the mountain-valley wind on the transportation of BC. Fig. 10 shows the vertical profile of the BC concentration along the QOMS's longitude of 86.95°E. During event A, high concentrations of BC appeared in the upper atmosphere of South Asia, and many BC aerosols were transported to most parts of the TP (Fig. 10a) due to the large-scale transport process. However, for events B-D, high concentrations of BC occurred along the southern slope of the Himalayas, and BC aerosols were only transported to a few areas on the northern slope of the Himalayas such as the Mt. Everest region (Fig. 10b-d) due to the local mountain-valley wind. As shown in Fig. S6, for events B-D, the up-valley wind on the southern side of the Himalayas can move BC aerosols up in the daytime, and the down-valley wind can cause the aerosols to descend in the Mt. Everest region at night.

To sum up, we found that the transport processes of BC aerosols from South Asia to the QOMS were different as the
seasons changed. In the monsoon season such as event A, BC aerosols were moved upward by the cyclone in the lower
atmosphere and were transported to QOMS by the southerly winds in the upper atmosphere. However, in the pre-monsoon
season such as events B-D, the mountain-valley wind played an important role in transporting the BC aerosols from the
southern slope of the Himalayas to the Mt. Everest region.
**4. Conclusions**
In this study, BC concentrations were measured from 15 May 2015 to 31 May 2017 at QOMS on the south edge of the
TP. Monthly, daily, and diurnal variations in BC concentrations were calculated to investigate the temporal characteristics and
potential sources of BC at QOMS. The results showed that the monthly mean BC concentrations reached the highest value in
the pre-monsoon season (923.1 $\pm$ 685.8 ng/m$^3$) and the lowest value in the monsoon season (88.5 $\pm$ 29.8 ng/m$^3$). The average
daily BC concentration was equal to 298.8 $\pm$ 341.3 ng/m$^3$, with a maximum in the pre-monsoon season (2772.3 ng/m$^3$) and a
minimum in the monsoon season (36.4 ng/m$^3$). For the diurnal variation in BC, there was an increase in the afternoon during
the non-monsoon periods, and high BC concentrations occurred from late night to morning, implying that the potential origin
of BC was from long-range transport. The BC concentrations remained low but peaked in the morning and in the noon during
the monsoon period, which might be due to local anthropogenic activities. In addition, the substantially high values of diurnal
variation in the BC concentrations in the pre-monsoon season suggest the high contributions of long-distance emissions carried
by mountain winds and glacier winds.
The seasonal cycle of BC concentrations at QOMS was closely correlated with the variation in the atmospheric circulation
and combustion emissions in South Asia. In the non-monsoon seasons, affected by the westerly winds, the air masses in the
Mt. Everest region were largely from Pakistan, northern Indian, and Nepal due to the high loading emissions from vegetation
fires. In the monsoon season, the southerly winds prevailed in the Mt. Everest region, and the air masses were mainly from the
Arabian Sea and the Bay of Bengal. Under intense precipitation scavenging of BC and extremely low levels of combustion
emissions in South Asia, the BC concentrations at QOMS were close to the background level in the monsoon season.
For the four heavy pollution episodes that occurred at QOMS with BC concentrations above 1000 ng/m$^3$, we found that
the transport processes of the BC aerosols from South Asia to the Mt. Everest region were different as the seasons changed. In
the monsoon season (using the pollution event during 8-10 June 2015 as an example), BC aerosols were efficiently driven
upward by the cyclone in the lower atmosphere in South Asia and transported to the Mt. Everest region by the southerly winds
in the upper atmosphere. However, during the pre-monsoon season (using the other three pollution events as examples), the
mountain-valley wind played an important role in transporting the BC aerosols across the Himalayas to the Mt. Everest region.
*Data availability.* All data are available upon requests made to the corresponding author.
*Competing interests.* The authors declare that they have no conflict of interest.
*Special issue statement.* This article is part of the special issue "Study of ozone, aerosols and radiation over the Tibetan Plateau
(SOAR-TP) (ACP/AMT inter-journal SI)". It is not associated with a conference.
*Acknowledgments.* This study was supported by the National Nature Science Foundation of China (41630754, 41675130,
41721091), Chinese Academy of Sciences (QYZDJ-SSW-DQC039), and State Key Laboratory of Cryospheric Science
(SKLCS-ZZ-2017). The authors thank the staff of the Qomolangma Atmospheric and Environmental Observation and
Research Station of Chinese Academy of Sciences for collecting data and the support of meteorological dataset. We also give
thanks to Tony Hansen for his suggestion on data processing and Xin Wan, Lekhendra Tripathee, and Yajun Liu for their help
to improve the quality of this paper.

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

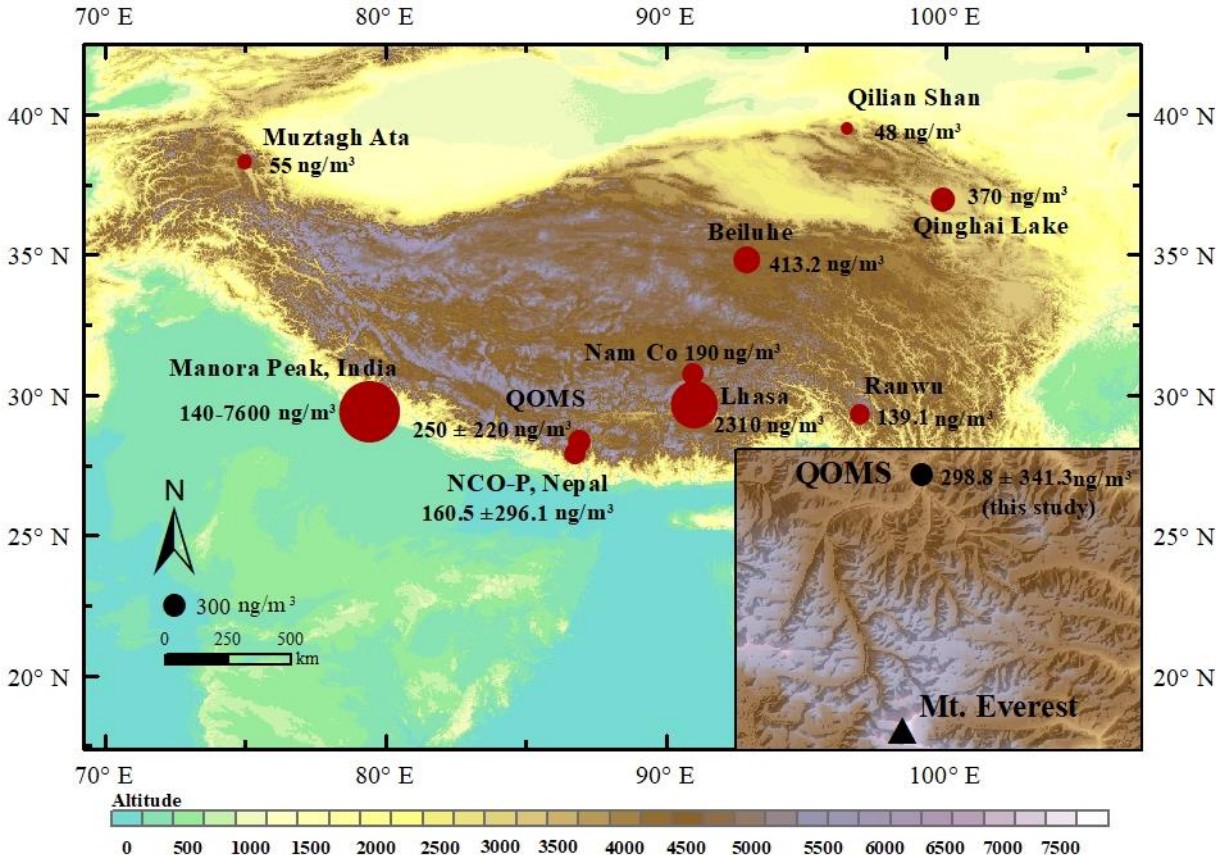

**Figure 1.** Distribution of BC concentrations over the TP based on the observed values at QOMS in this study (black circle) and from previous studies (red circles), i.e., at QOMS (Cong et al., 2015a), Nam Co (Wan et al., 2015), Lhasa (Li et al., 2016b), Ranwu (Wang et al., 2016), Qilian Shan (Zhao et al., 2012), Beiluhe (Wang et al., 2016), Qinghai Lake (Li et al., 2013), Muztagh Ata (Cao et al., 2009), Manora Peak, India (Ram et al., 2010), and NCO-P, Nepal (Marinoni et al., 2010).

609

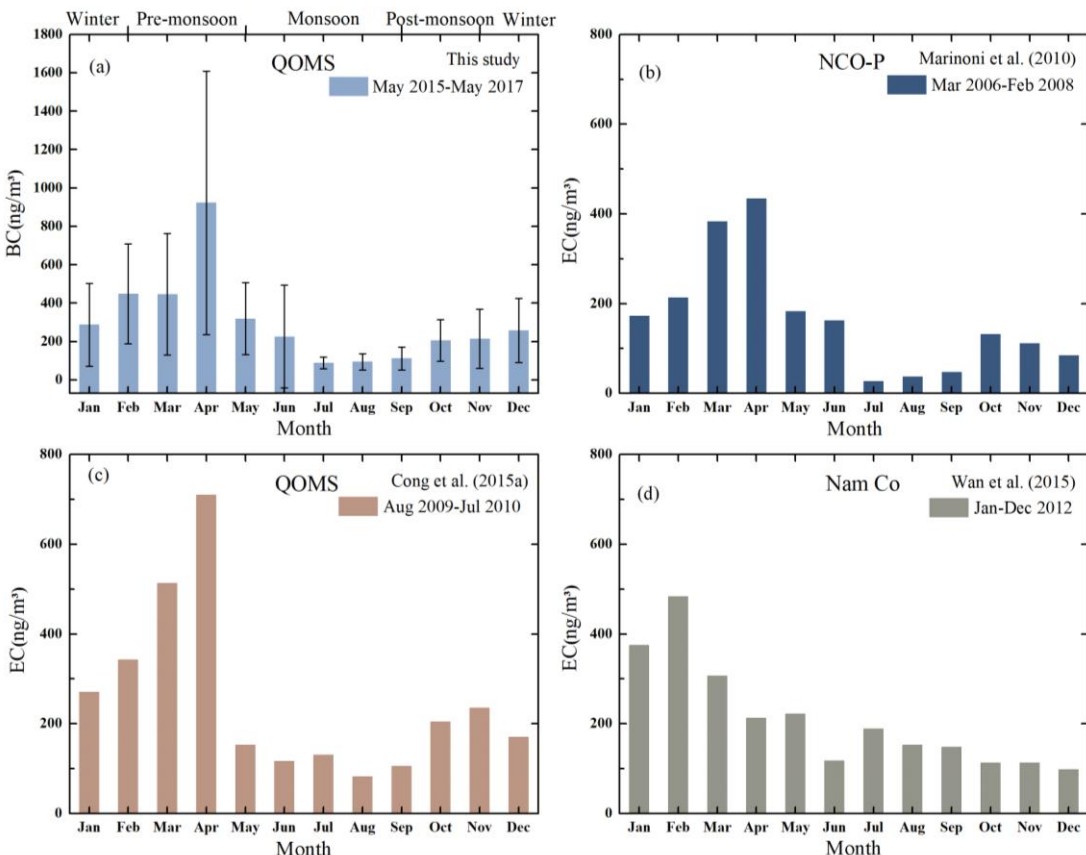

610

**Figure 2. (a) Monthly mean BC concentrations at QOMS from May 2015 to May 2017 in this study; (b) Monthly mean EC at NCO-P from March 2006 to February 2008 from Marinoni et al. (2010); (c) Monthly mean EC at QOMS from August 2009 to July 2010 from Cong et al. (2015a); (d) Monthly mean EC at Nam Co station from January to December during 2012 from Wan et al. (2015).**

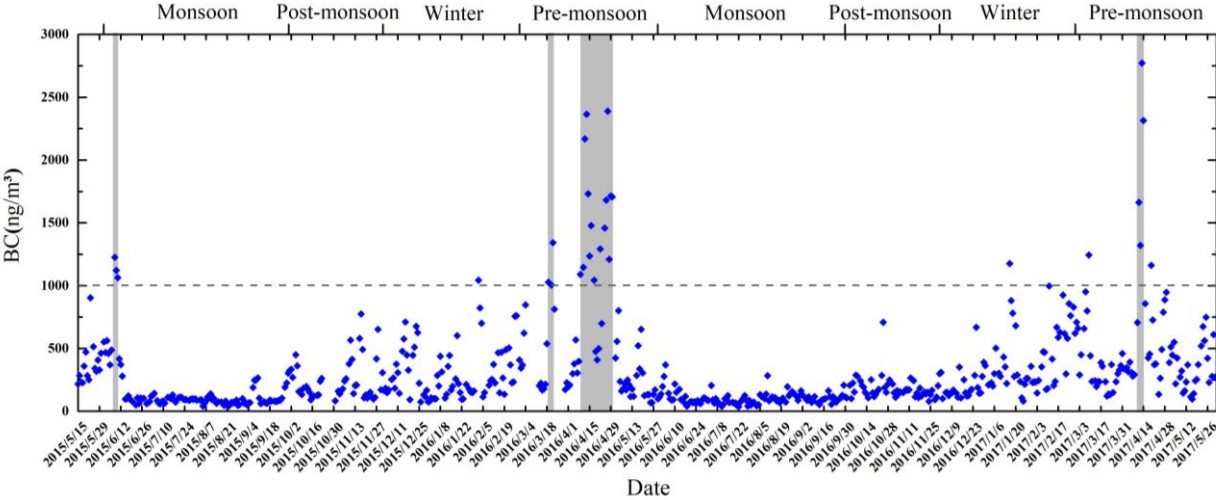


**Figure 3. Daily mean BC concentrations at QOMS during study period (the gray bars represent the continuous high values more**
**than 1000 ng/m³).**

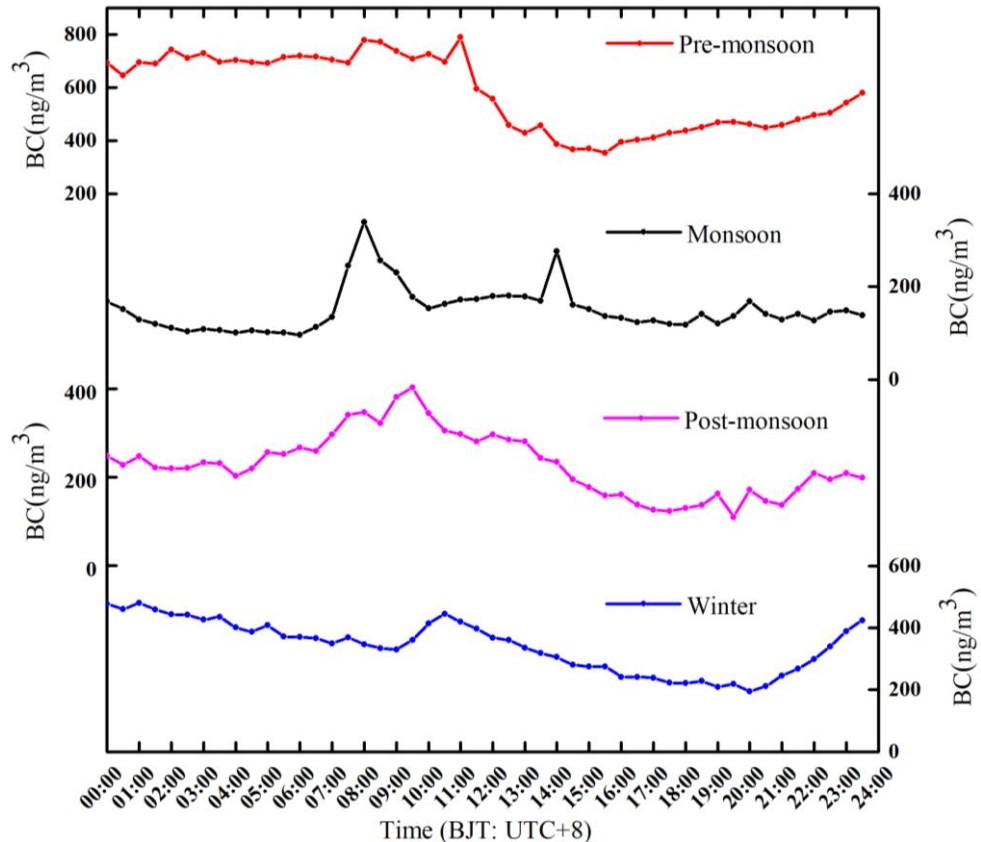


**Figure 4. Diurnal variation in BC concentrations (every half an hour) at QOMS during study period.**

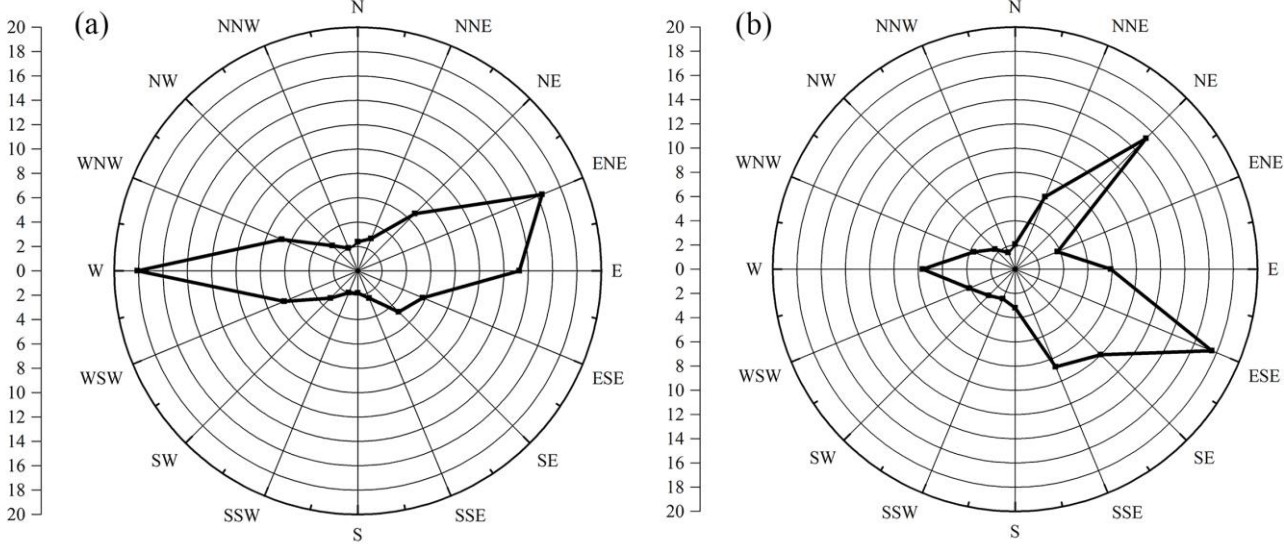


**Figure 5. Wind direction frequency at QOMS in the pre-monsoon season (a) 00:00-12:00 BJT; (b) 12:00-24:00 BJT.**

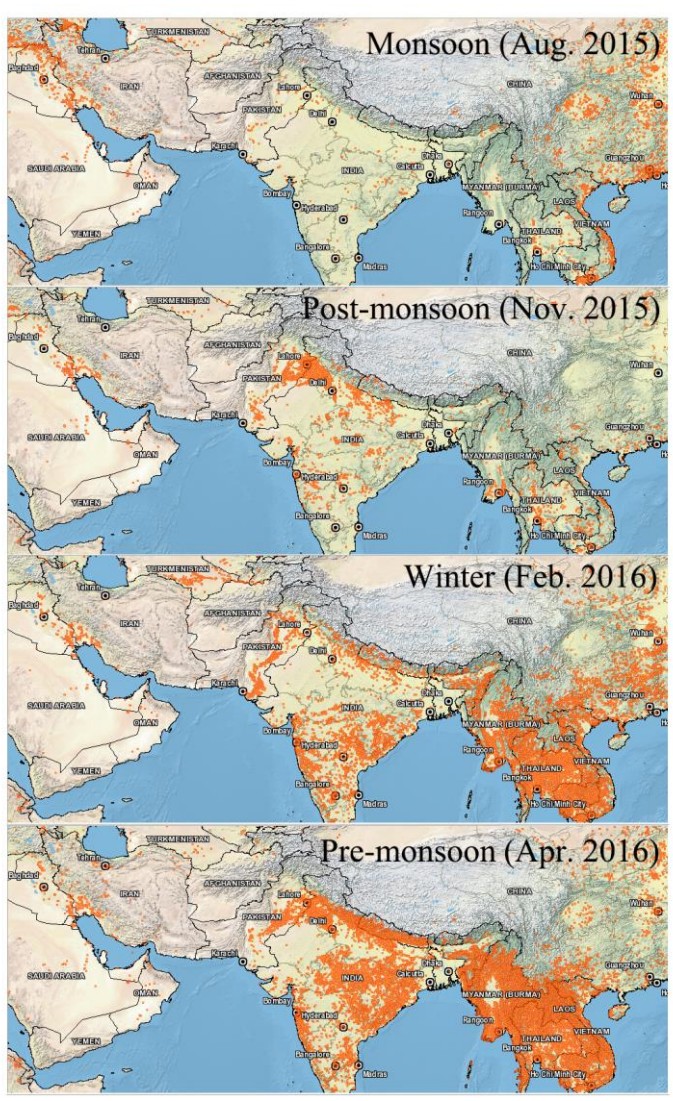


**Figure 6. Distribution of fire spots in different seasons from August 2015 to April 2016.**


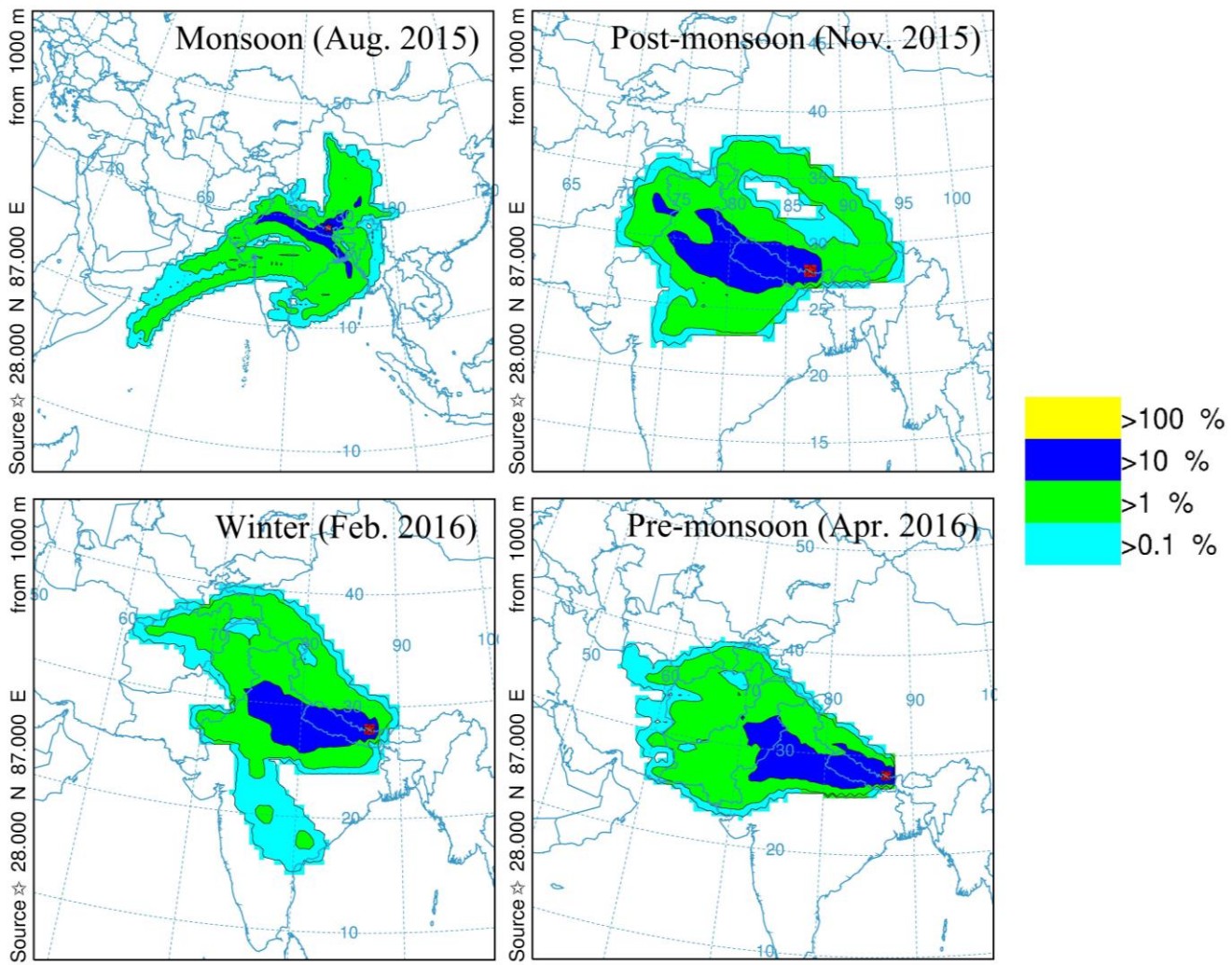

**Figure 7. Frequency plots for 5-day backward trajectories calculated by HYSPLIT model at QOMS in different seasons from August 2015 to April 2016.**

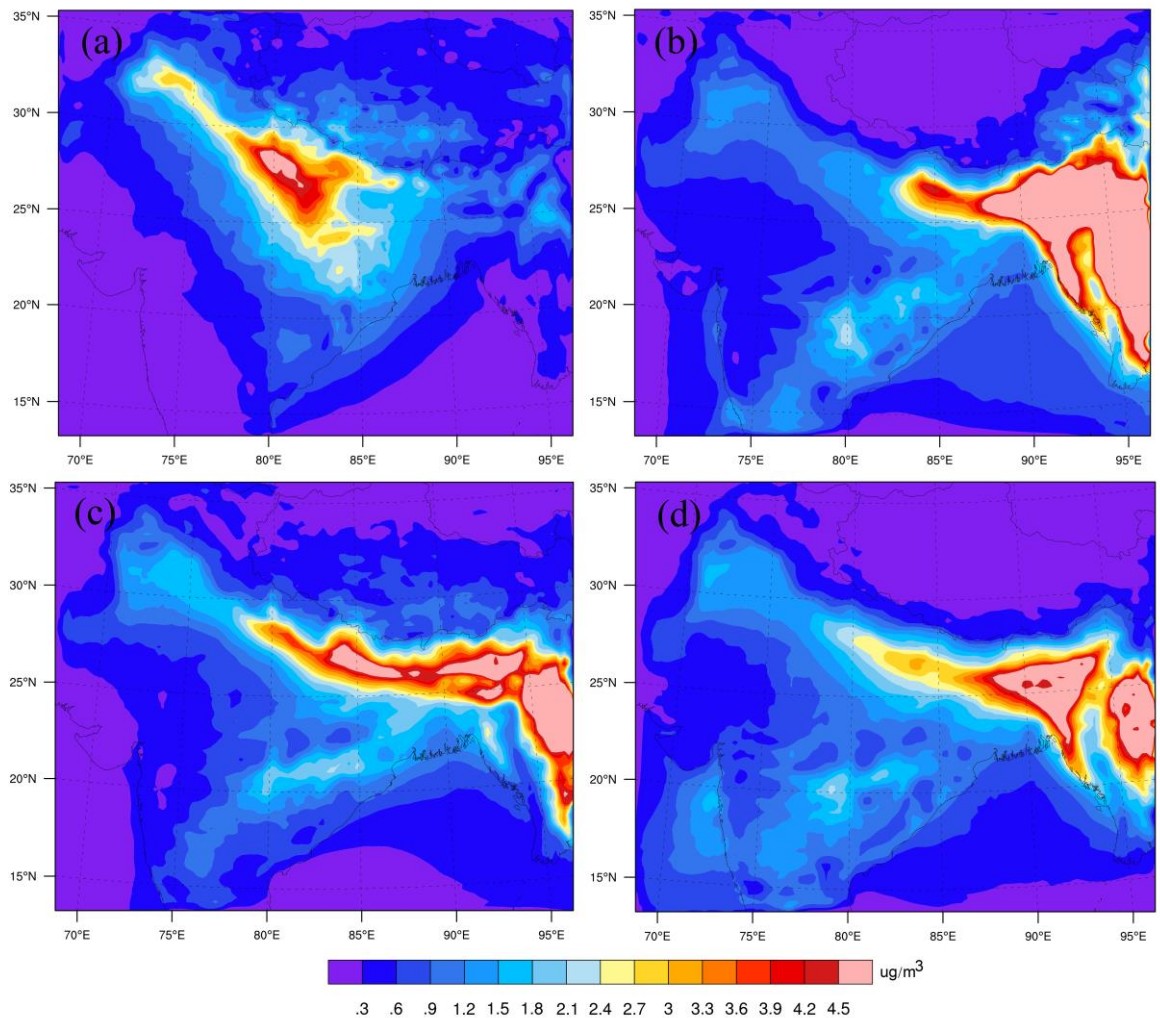

632

Figure 8. Mean BC concentration simulated by WRF-Chem model at QOMS and its vicinities: (a) event A; (b) event B; (c) event C;
(d) event D.

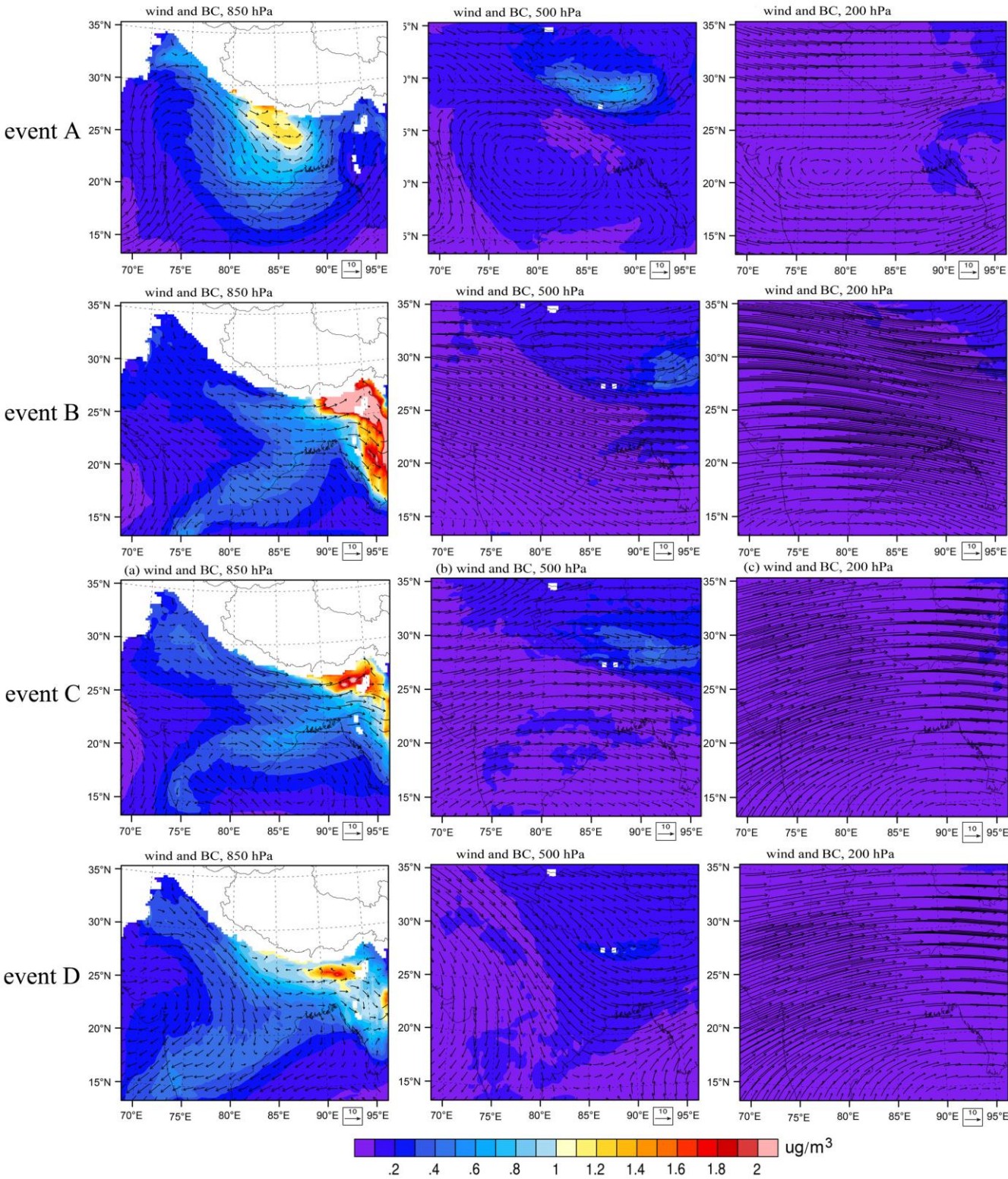

**Figure 9. Mean BC concentration and wind at 850 hPa, 500 hPa, and 200 hPa simulated by WRF-Chem model at QOMS and its vicinities: event A (the first row); event B (the second row); event C (the third row); event D (the last row).**

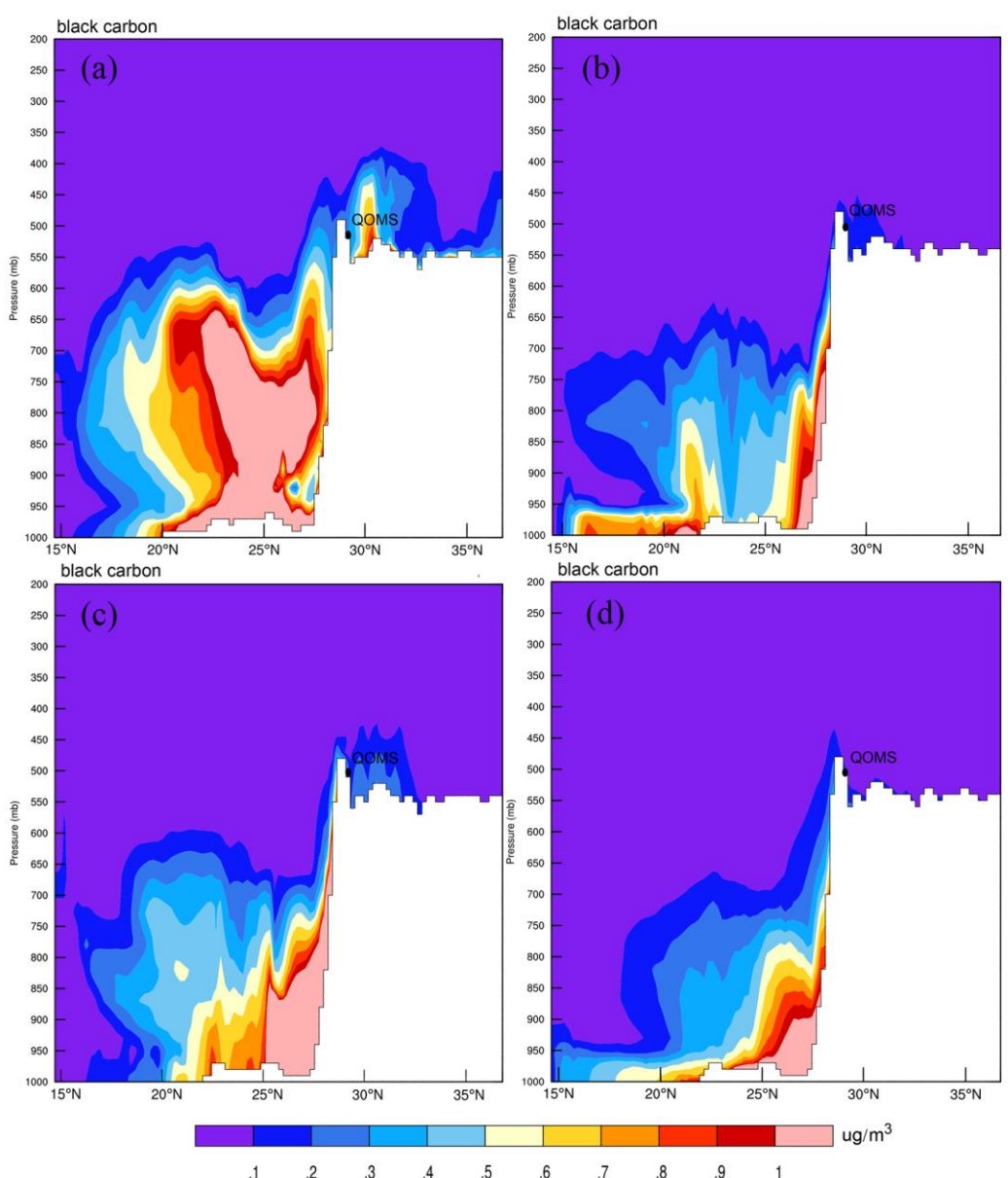


**Figure 10. Vertical profiles of mean BC concentration along the QOMS's longitude of 86.95°E: (a) event A; (b) event B; (c) event C;**
**(d) event D.**
