# Peer review of "Concentration, temporal variation and sources of black carbon in the Mount Everest region retrieved by real-time observation and simulation"

_Atmospheric Chemistry and Physics, 2018_

## Referee Comment (RC1) · Anonymous Referee #1 · 15 Apr 2018

The manuscript investigated the seasonal and diurnal variations of BC and its potential source regions in the Tibetan Plateau. It is significant to research the effect of pollutant aerosol on the Tibetan Plateau. But there are some comments to improve the manuscript. 1. Merge Table 1 in Figure 1. So the reader can directly see the distribution of BC in the Tibetan Plateau. 2. Figure 2 and Figure 6 are redundant. The weather data is not significant in the research. 3. Add some important references to compare with the results, e.g. Xin et al., BAMS, 2015. 4. Compare the BC concentration of WRF-Chem with the observation in the Tibetan Plateau.

[Figure]

2018.

---

## Referee Comment (RC2) · Anonymous Referee #2 · 18 Apr 2018

The authors conducted a detailed analysis of the BC concentration measurements at the Qomolangma station. The measurement, with a high temporal resolution and a relatively long period, provides very valuable information for the understanding of BC sources and transport to the Himalayas. The authors further combined observations with model simulations to investigate the BC transport mechanism. The analysis is comprehensive and the manuscript is generally well written. Before it can be considered for publication, I have a few comments and suggestions.

1. Section 2.2: Since different measurement methods may lead to quite different BC or EC concentrations, I suggest adding some discussions in this section on the possible

difference in measured BC concentrations between the AE-33 used in this study and some widely-used methods from previous studies (e.g., thermal-optics method, SP2, etc.).

2. Section 3.4: The authors did a detailed analysis on possible BC sources and transport mechanisms for four pollution events, which is great. However, the evaluation of WRF-Chem model simulation seems missing here. Without knowing the model performance, it is difficult to be convinced by the source and transport analysis of model results. At least, the authors could compare modeled BC concentrations at this site with their observations. If possible, the modeled wind and precipitation can be also evaluated against some reanalysis or satellite products. If this takes too much time, the authors could also cite and discuss some previous studies where the WRF-Chem simulations have been evaluated in the TP and surrounding regions.

3. Line 15: "... concentrations were significantly greater from mid-night to noon. . ." This sentence is a little confusing. Do you mean "concentrations increased from mid-night to noon"?

4. Line 16: ". . ., implying the potential contribution from the long-range transport." It is not very straightforward for readers to understand why such diurnal variation implies the contribution from the long-range transport. Could you please rephrase the sentence and clarify the point?

5. Line 40: For the authors' information, a recent study (Lee et al., 2017) investigated BC deposition effects on reducing snow albedo over the Tibetan plateau based on satellite observation analysis. This study can be cited here.

Lee, W. L., et al. (2017): Impact of absorbing aerosol deposition on snow albedo reduction over the southern Tibetan plateau based on satellite observations, Theor. Appl. Climatol., 129, 1373-1382, doi:10.1007/s00704-016-1860-4

6. Line 57: For the authors' information, a recent study (He et al., 2014b) has also

used a global CTM to investigate the sources of BC over the Tibetan Plateau based on a tag-tracer technique, which can be cited here.

He, C., et al. (2014b): Black Carbon Radiative Forcing over the Tibetan Plateau. Geophy. Res. Lett., 41, 7806–7813, doi:10.1002/2014GL062191

7. Lines 102–112: It would be more informative if the authors could provide the uncertainty/accuracy associated with this algorithm for BC concentration calculation.

8. Section 2.3 "Model simulation": A number of studies (e.g., Flanner et al., 2007; Liou et al., 2014; He et al., 2017) have shown significant effects from BC in snow on albedo simulations. This albedo effect and feedback may exert an important impact on model simulations. Did the authors include such "dirty snow" effect in the WRF-Chem simulations? I suggest adding a brief discussion on this issue.

Flanner, M. G., et al. (2007), Present-day climate forcing and response from black carbon in snow, J. Geophys. Res., 112, D11202, doi:10.1029/2006JD008003

He, C., et al. (2017): Impact of Snow Grain Shape and Black Carbon-Snow Internal Mixing on Snow Optical Properties: Parameterizations for climate models, J. Climate, 30(24), 10019-10036, doi:10.1175/JCLI-D-17-0300.1

Liou, K. N., et al. (2014), Stochastic parameterization for light absorption by internally mixed BC/dust in snow grains for application to climate models, J. Geophys. Res.-Atmos., 119, 7616–7632, doi:10.1002/2014JD021665

9. Line 183: "..., which might be owing to the surrounding local emissions." Is there any reference/observation showing surrounding emissions? Is there any populated city or town around the observational site? More information would better convince readers.
* * *

---

## Referee Comment (RC3) · Anonymous Referee #3 · 26 Apr 2018

This paper is talking about mass concentrations, temporal variations and source regions along with transported mechanisms of airborne BC observed in Mountain Everest region. The authors have revised this paper followed by the reviewer's comments. I suggest that this paper can be published to ACP and some minor comments are listed as: 1. Line 16, change "...in the non-monsoon season" to "...during the monsoon season". 2. Line 78-79. What is the time resolution for measurements in meteorological parameters? The authors should state. 3. Line 129-133. The authors should cite papers for utilizations of HYSPLIT model. 4. Line 141. Please replace "........at QQMS (Fig. 3c) (Cong et al., 2015a)" by "........at QQMS from August 2009 to July 2010 (Fig. 3c) (Cong et al., 2015a)". 5. Line 161. I think the decreased BC during

the monsoon is due to washout by precipitation. So, please remove humidity in the sentence of "….increasing humidity and…." 6. Fig. 11. It is better to point out the location of QQMS in all panels of this figure for the reader to follow it.

---

## Referee Comment (RC4) · Anonymous Referee #4 · 30 Apr 2018

1. This manuscript present analysis of the high-resolution measurement of black carbon (BC) at Qomolangma (Everest) station of Chinese Academy of Sciences during 15 May 2015 to 31 May 2017, together with model simulations to investigate the possible transport mechanisms of BC. Generally, the manuscript is well organized, but many sentences and even paragraphs still need to be clarified or improved. Though I have marked many places in the text, I believe that there are still more problematic phases or sentences to be identified and corrected. I suggest that the whole text should be carefully checked and improved with the help of an English editor.

2. Some of the explanations are not convincing. For example, in line 182 of page

6, it reads "The valley wind from north in the morning, could bring the short-distance emissions from local cooking or heating to QOMS. BC concentrations appeared two peaks in the morning and after the noon in the monsoon season, which might be owing to the surrounding local emission." Why it occurred in the morning and afternoon in the monsoon season, not other times and in other seasons? It should be clarified to what extent the daily and seasonal values and patterns obtained in this study are influenced by local emissions.

3. Section 3.2 is not well written. What do you want to say through these comparisons?

4. The authors should indicate what is new in this study. It seems to me that most of the results are similar to those obtained in previous studies, although different instruments and models might be used in different studies.

5. More corrections and comments are marked in the text.

Please also note the supplement to this comment:
https://www.atmos-chem-phys-discuss.net/acp-2018-183/acp-2018-183-RC4-supplement.pdf

**Supplement:**

**Concentration, temporal variation and sources of black carbon in the Mount Everest region retrieved by real-time observation and simulation**

Xintong Chen [1, 4], Shichang Kang [1, 2, 4], Zhiyuan Cong[2, 3], Junhua Yang [1], Yaoming Ma [3]

[1]State Key Laboratory of Cryospheric Science, Northwest Institute of Eco-Environment and Resources, Chinese Academy of Sciences, Lanzhou 730000, China

[2]CAS Center for Excellence in Tibetan Plateau Earth Sciences, Chinese Academy of Sciences, Beijing, 100101, China

[3]Key Laboratory of Tibetan Environment Changes and Land Surface Processes, Institute of Tibetan Plateau Research, Chinese Academy of Sciences, Beijing 100101, China

[4]University of Chinese Academy of Sciences, Beijing 100049, China.

*Correspondence to*: Shichang Kang (shichang.kang@lzb.ac.cn)

**Abstract.** Based on the high-resolution measurement of black carbon (BC) at Qomolangma (Everest) station of Chinese Academy of Sciences during 15 May 2015 to 31 May 2017, we investigated the seasonal and diurnal variations of BC and its potential source regions. Monthly and daily mean BC concentrations reached the highest values in the pre-monsoon season which are at least one magnitude higher than the lowest values in the monsoon season. For the diurnal variation, BC concentrations were significantly greater from mid-night to noon in the pre-monsoon season and showed increasing trend in the afternoon in the non-monsoon seasons, implying the potential contribution from the long-range transport. In the monsoon season, BC concentrations appeared two peaks in the morning and after the noon might be affected by the local anthropogenic activities. By analyzing the simulation results from the backward air-mass trajectories and the fire spots distribution from the MODIS data, we found that the seasonal cycle of BC was significantly influenced by atmospheric circulation and combustion intensity in the Mt. Everest region. The transport mechanisms of BC were revealed using WRF-Chem simulation during severe pollution episodes. For the pollution event in the monsoon season, BC aerosols in South Asia could be uplifted and transported to the Mt. Everest region by the southward winds in the upper atmosphere. However, for the events in the pre-monsoon season, BC from northern India was brought and concentrated in the southern slope of the Himalayas by the northwesterly winds in the lower atmosphere and then transported across the Himalayas by the mountain-valley wind, while relatively less BC from northwestern India and Central Asia could be transported to the Mt. Everest region by the westerly winds in the upper atmosphere.

**1 Introduction**

Black Carbon (BC), from the incomplete combustion of fossil fuels or biomass burning, has drawn lots of attention due to its influences on environment and human health (Anenberg et al., 2012; Bond, 2004; Ramanathan et al., 2005), and is seen as an important factor that may lead to global warming besides greenhouse gases (Hansen et al., 2000; Jacobson, 2002; Bond et al., 2013). BC can greatly absorb solar radiation and causes atmosphere heating (Jacobson, 2001; Ramanathan et al., 2005;

Ji et al., 2015). Moreover, BC as fine particles can suspend in the atmosphere for about one week and transported away from its emission sources, and be removed by dry and wet deposition (Oshima et al., 2012; Cooke et al., 2002; Jurado et al., 2008). When BC deposited on snow and ice, it can significantly reduce surface albedo and accelerate glacier and snow cover melting, causing the impact on regional climate, hydrology, and water resources (Ming et al., 2008; Li et al., 2018; Ramanathan and Carmichael, 2008).

The Tibetan Plateau (TP), generally known as the "Third Pole", is the highest plateau with a large number of glaciers and snow cover (Kang et al., 2010; Lu et al., 2010; Yao et al., 2012). Even though the TP is remote region with little affected by anthropogenic activities, previous observations have indicated that BC is an important contributor to the rapid shrinking of glaciers over the TP via decreasing surface albedo and atmospheric warming (Xu et al., 2009; Yang et al., 2015; Yasunari et al., 2010; Li et al., 2017; Zhang et al., 2017b; Qu et al., 2014; Ji, 2016; Xu et al., 2016). Moreover, previous studies have also suggested that the emissions from South Asia and East Asia are the major sources of BC in the TP (Lu et al., 2012; He et al., 2014; Zhang et al., 2015; Yang et al., 2018), and the high emissions from South Asia can across the Himalayas and further transport to the inland of the TP (Luthi et al., 2015; Xu et al., 2014; Cong et al., 2015a; Kang et al., 2016; Wan et al., 2015). Meanwhile, seasonality of BC aerosols are closely related to atmospheric circulation that helps to bring the BC aerosols across the Himalayas (Cong et al., 2015a; Yang et al., 2018; Zou et al., 2008). Additionally, a large number of studies have also demonstrated that the BC and dust from Central Asia and Northern Africa could also be transported to the TP (Wang et al., 2016; Lu et al., 2012; Zhao et al., 2012; Wu et al., 2010; Zhang et al., 2015).

Mt. Everest could be regarded as the very sensitive area under the influence of BC aerosols. Previous research on atmospheric BC in the Mt. Everest region was mainly based on thermal/optical analytical method, using quartz filter samples (Cong et al., 2015a). However, investigating in diurnal and seasonal variations of BC still lacks in the region. Therefore, to fulfill the gaps and understand the variations and sources of BC in the pristine region, there is a need for an efficient approach and studies. The aethalometer can provide high-resolution and continuous real-time observation data on BC concentration which is very important and necessary to better depict the characteristics of BC and its effects on the environmental change.

In comparison with the observation, the numerical models can better represent the atmospheric physical and chemical processes. Many studies have used global climate models (GCMs) and chemical transport models (CTMs) to investigate the origin and transportation of BC over the TP (Lu et al., 2012; Zhang et al., 2015; Menon et al., 2010; Kopacz et al., 2011). However, due to the coarse resolution, it is difficult for the CTMs and GCMs to capture the surface details of the TP (Ji et al., 2015; Gao et al., 2008). Regional climate models (RCMs) can compensate for the shortcomings of coarser global model grids by the high-resolution simulations. In the recent decades, RCMs have been developed to include multiple modules and atmospheric chemistry processes. Also the advanced regional climate-chemistry model, Weather Research and Forecasting (WRF) model (Skamarock et al., 2005); with Chemistry (WRF-Chem), has been successfully applied for air quality research in the TP region (Yang et al., 2017; Yang et al., 2018).

Here, we present real-time data of BC concentration measured by the new generation aethalometer (AE-33), from 15 May 2015 to 31 May 2017. The observed results are used to characterize the temporal variation and provide the solid information on the possible sources and transport mechanism of BC. Combined high-resolution measurement of BC concentration and WRF-Chem model we investigated the concentration level, temporal variation, and sources of BC in the Mt. Everest region. The purpose of this study is to understand the impact of trans-boundary atmospheric BC on the Mt. Everest region and depict the transport pathways of BC in different spatiotemporal scales.

**2 Materials and methods**

**2.1 Sampling site and meteorological condition**

Mt. Everest (27.98°N, 86.92°E, 8844 m a.s.l.), the summit of the world, is located in the central Himalayas. The southern slope of the Mt. Everest is adjacent to the Indian continent, and the climate is warm and humid under the influence of the Indian summer monsoon. Conversely, the northern side is cold and dry since the warm and humid airflow cannot reach there. The Qomolangma Atmospheric and Environmental Observation and Research Station (QOMS, 28.36°N, 86.95°E, 4276 m a.s.l.) of Chinese Academy of Sciences (Fig. 1) is located in the northern slope of the Mt. Everest, which was established for the continuous monitoring of atmospheric environment (Cong et al., 2015a; Ma et al., 2011).

The meteorological parameters, i.e., air temperature, air pressure, humidity, wind speeds and wind direction, were recorded by an automatic weather station at QOMS. Meanwhile, the precipitation data were collected by artificial measurement, as shown in Fig. 2. The entire year was divided into four seasons according to the Indian monsoon transitions characteristics, which includes pre-monsoon (March to May), monsoon (June to September), post-monsoon (October to November), and winter (December to February) (Praveen et al., 2012; Zhang et al., 2017a). A clear seasonal cycle of temperature and humidity could be observed from Fig. 2. Specifically, higher temperature was observed during the monsoon season while lower during winter, with a maximum in July and a minimum in January. Humidity followed the similar trend, with higher values from late July to early August and lower values from December to February. During the observation period, the wind speed increased significantly from November to April. The wind direction at QOMS is affected by the local topography that with a series of small valleys. During the pre-monsoon season (Dry-period), the westerly and southerly winds begin to develop and play the important role in atmospheric pollution circulation. However, during the monsoon season, southwesterly winds prevail and bring much moisture from the Indian Ocean to the Mt. Everest region increasing the humidity and precipitation. With the retreat of monsoon, southwesterly winds decrease and the popular wind direction changes to westerly and northeasterly in winter with limited moisture (Fig. 2).

**2.2 BC measurements**

The airborne BC concentrations at QOMS were monitored by the "Next Generation" aethalometer model AE-33 (Magee Scientific Corporation, USA). The instrument was set in an indoor room with inlet installed at about 3 m above the ground level and was operated at an airflow rate of 4 LPM with 1 min time resolution at QOMS.

AE-33 has seven fixed wavelengths (i.e., 370，470，520，590，660，880 and 950 nm), which can acquire BC concentration according to light absorption and attenuation characteristics from the different wavelengths (Hansen et al., 1984; Drinovec et al., 2014). Generally, BC concentration measured at 880 nm is used as the actual BC concentration in the atmosphere, as the absorption of other species of aerosols is greatly reduced in this wavelength (Sandradewi et al., 2008a; Sandradewi et al., 2008b; Fialho et al., 2005; Yang et al., 2009; Drinovec et al., 2014). Compared to previous aethalometers, AE-33 uses doulspot method and a real-time calculation of the "loading compensation parameter", which can compensate for the "spot loading effect" and obtain high quality BC concentration (Drinovec et al., 2014). The algorithm is as follows:

$$\mathrm{BC\ (reported)}=\mathrm{BC(zero\ loading)}\times(1-k\mathrm{ATN}) \quad (1)$$

$$\mathrm{ATN}=-100\ln(I/I_0) \quad (2)$$

$$\mathrm{BC1}=\mathrm{BC}\times(1-k\mathrm{ATN1}) \quad (3)$$

$$\mathrm{BC2}=\mathrm{BC}\times(1-k\mathrm{ATN2}) \quad (4)$$

Where, BC (reported) is the non-compensated BC concentration; BC (zero loading) is the desired ambient BC value that would be obtained in the absence of any loading effect; k is the loading compensation parameter; I and $I_0$ are the light intensity of the measurement spot and reference spot; ATN is the attenuation coefficient. BC component of aerosols is analyzed on two parallel spots drawn from the same input stream in AE-33, but collected at different rates of accumulation. It means that we can obtain different ATN but the same loading parameter k (Drinovec et al., 2014). Combining Eq. (3) and Eq. (4), the "loading compensation parameter" k and the desired value of BC compensated back to zero loading can be calculated.

**2.3 Model simulation and datasets**

The WRF-Chem version 3.6 was used to analyze the spatial distribution, transport mechanism, and source apportionment of BC during the four observed pollution episodes. The WRF-Chem model is the expansion of WRF meteorological model and considers complex physical and chemical process such as the emission and deposition, advection and diffusion, gaseous and aqueous chemical transformation, and aerosol chemistry and dynamics (Grell et al., 2005). Here, the numerical experiments were performed at 25 km horizontal resolution with 122 and 101 grid cell in the west-east direction and north-south, respectively. The simulated domain was centered at 25°N, 82.5°E and had a 30-layer structure with the top pressure of 50 hPa. Key physical and chemical parameterization options for the WRF-Chem modeling were according to the previous study in the TP (Yang et al., 2018). The initial meteorological fields were from National Centers for Environmental Prediction (NECP) reanalysis data with a horizontal resolution of 1 ° × 1 ° at 6-h time intervals. The anthropogenic emission inventory was obtained from the Intercontinental Chemical Transport Experiment-Phase B (INTEX-B) (Zhang et al., 2009) with a

resolution of 0.5° × 0.5°. The biogenic emissions were from the Model of Emission of Gases and Aerosol from Nature (MEGAN), and the fire emissions inventory was based on the fire inventory from NCAR (FINN) (Wiedinmyer et al., 2011). Additionally, the Model for Ozone and Related chemical Tracers (MOZART, http://www.acom.ucar.edu/wrf-chem/mozart.shtml) (Emmons et al., 2010) dataset were used to create improved initial and boundary conditions for BC simulations during these pollution episodes.

Furthermore, predicting the source region of BC, we used HYSPLIT-4 model to calculate the backward trajectories of air masses, and the data for calculation were from National Centers for Environmental Prediction/National Center for Atmospheric Research（NCEP/NCAR）data (2.5 ° × 2.5 °, 17 vertical levels). The active fire product provided by Fire Information for Resource Management System (FIRMS, https://firms.modaps.eosdis.nasa.gov/firemap/), was chosen to investigate the biomass burning emissions over the region in different seasons.

**3. Results and discussion**

**3.1 Temporal variations of BC**

**3.1.1 Monthly variation of BC**

Monthly mean BC concentrations at QOMS is shown in Fig. 3a. There was a significant increase in BC concentrations in winter and the highest value occurred during the pre-monsoon season (923.1 ± 685.8 ng/m$^3$ in April). Meanwhile, during monsoon, the lower BC concentrations were recorded and the lowest value was observed in July (88.5 ± 29.8 ng/m$^3$). This seasonal change was consistent with the previous studies of element carbon (EC or BC) at Nepal Climate Observatory-Pyramid station (NCO-P, 27.95°N, 86.82°E, 5079m a.s.l.) (Fig. 3b) (Marinoni et al., 2010) and at QOMS (Fig. 3c) (Cong et al., 2015a), indicating that the similar BC source between the southern and northern sides of the Himalayas. As the EC was sampled by the quartz filters and detected using the thermal/optical analytical method in the previous studies, there may have been some disparities in the values of EC with these of BC. The monthly variation of EC at Nam Co Monitoring and Research Station for Multisphere Interactions (Nam Co station, 30.46°N, 90.59°E, 4730 m a.s.l.) (Fig. 3d) (Wan et al., 2015) also showed the similar variation, but the peak value of EC occurred in winter. Additionally, monthly mean EC concentrations at Nam Co station were generally lower than that at QOMS, suggesting that anthropogenic activities impacts in the inland TP were weaker than that in the south edge of the TP. Previous studies have demonstrated that the influence of polluted air masses from the "Atmospheric Brown Clouds" over South Asia could reach to the southern foothills of the Himalayas, and the mountain-valley breeze circulation carried the polluted air masses onto the TP (Luthi et al., 2015; Cong et al., 2015a; Bonasoni et al., 2008; Yang et al., 2018). Therefore, the seasonal cycle of BC concentrations at QOMS was likely affected by the atmospheric circulation and the emissions from South Asia, and these will be further explained in Section 3.3.

**3.1.2 Daily variation of BC**

Fig. 4 shows the daily mean BC concentrations at QOMS which presents a significant seasonal pattern, with a maximum during the pre-monsoon season (2772.3 ng/m$^3$) and a minimum during the monsoon season (36.4 ng/m$^3$). During the monsoon season, BC concentration was observed to be lower than 150 ng/m$^3$, but it gradually increased during the post-monsoon and winter. The mean concentration of daily BC at QOMS was 298.8 ± 341.3 ng/m$^3$, which was close to the previous result (250 ± 220 ng/m$^3$) (Cong et al., 2015a). Such a result demonstrates that our results are consistent with the previous finding and systematic sampling of aerosol is also important to obtain BC values in the region.

The comparison between daily mean BC concentrations (Fig. 4) and the meteorological parameters (Fig. 2) suggested that the increasing humidity and precipitation during monsoon led to washout of atmospheric particles promoting the wet deposition of BC. This process caused a decrease in BC concentrations during monsoon representing the background level during the period. The prevailing wind direction during the monsoon period was southwesterly while in non-monsoon was dominated by westerly. Therefore, the variations of BC might be linked to the influence of meteorological conditions and the contribution of long-distance transport from urbanized areas to QOMS. Moreover, it cannot be ignored that there were continuous high concentrations of BC above 1000 ng/m$^3$ during 8-10 June 2015, 19-22 March 2016, 9-18 April 2016, and 11-14 April 2017, indicating the heavy pollution episodes happened at QOMS during those days. The detailed analysis for these pollution events is presented in Section 3.4.

**3.1.3 Diurnal variation of BC**

Diurnal variation characteristics can be used to analyze the impact of local meteorological process and anthropogenic activities on BC concentrations at QOMS. The half-hourly mean BC concentrations are presented in Fig. 5. The diurnal BC concentrations in the pre-monsoon season were significantly higher than those in other seasons, and remained high values from mid-night to noon and increased gradually after the lowest value around 15:00. A similar increasing trend for BC was observed in the afternoon mostly during the post-monsoon and winter periods, and highest BC concentration occurred from mid-night to noon. During the monsoon season, the BC concentrations remained low values with two peaks in the morning and after the noon respectively. Previous studies have demonstrated that the local wind system in the northern slope of the Mt. Everest were composed by a morning "valley wind", a late morning-afternoon "glacier wind" weakened by "valley wind", and an evening-early night "mountain wind"(Zou et al., 2008). The QOMS is located in the s-shape valley north of the Mt. Everest (Ma et al., 2011). The down mountain wind or glacier wind from south developed in the afternoon and at night, which provided the potential possibility for pollutants from long-distance source transported to QOMS along the valley and enabled the increase of BC concentrations in the non-monsoon periods. The valley wind from north in the morning, could bring the short-distance emissions from local cooking or heating to QOMS. BC concentrations appeared two peaks in the morning and after the noon in the monsoon season, which might be owing to the surrounding local emissions.

To explain the significant high values during mid-night to noon in the pre-monsoon season, the wind direction frequency

at QOMS during 0:00-12:00 and 12:00-24:00 are presented in Fig. 6. During the sampling period in the pre-monsoon season, W (west) winds prevailed from mid-night to noon (Fig. 6a), accounting for 18.1% of the total wind directions, followed by ENE (east-northeast) winds (16.4%). It is consistent with the discussion above that there exists potential impacts on BC concentrations at QOMS from long-distance human activity emissions, which can be carried by westerly winds, i.e., down mountain winds (Cong et al., 2015b). Moreover, the WRF-Chem simulation results showed that, the profile of equivalent potential temperature (EPT) increased with altitude and the planetary boundary layer height (PBLH) and wind speed were much lower from mid-night to noon (Fig. S1), indicating a more stable atmosphere that obstructs the diffusion of BC aerosols. While ESE (east-southeast) and NE (northeast) winds prevailed from noon to mid-night (Fig. 6b), accounting for 17.6% and 15.3% of total wind directions, respectively. Several villages are located easterly (around 5 km away) from QOMS thus lower BC values might reflect the short-distance emission sources.

**3.2 Comparison of BC concentrations with other sites in the TP**

In order to better understand the BC loading level, we compared our results with previous studies from other locations over the TP. As listed in Table 1, BC concentrations on Muztagh Ata Mountain and Qilian Mountain presented lower values, which can be regarded as the background concentration level for inland Asia (Zhao et al., 2012; Cao et al., 2009). In contrast, observed BC concentrations at QOMS were relatively higher than other remote locations in the TP, which has an increasing trend from the southern edge of the TP to the inland TP. Such as Nam Co and Ranwu, sites are isolated from anthropogenic activities with relatively clean atmospheric environments, but BC concentration at these two sites is recorded up to 130 ng/m$^3$, which may be likely due to the influence of long-range transport from South Asia (Wan et al., 2015; Wang et al., 2016). Compared with locations in the southern slope of the Himalayas (e.g., NCO-P and Manora Peak), the BC concentration at QOMS was close to that at NCO-P, but much lower than that at Manora Peak, which is near to the polluted areas in South Asia, and largely affected by anthropogenic emissions (Marinoni et al., 2010; Ram et al., 2010). It implied that combustion emissions from South Asia not only can affect the lower latitudes in the vicinity, but also can be transported to the higher latitudes in the Himalayas and even to the TP. However, the BC concentration at Lhasa city was higher than other remote sites in the TP, which was mainly from the local vehicle emissions (Li et al., 2016). But for BC concentration at Qinghai Lake, it was higher than that at the other south sites in the TP, because of the long-range transport of higher anthropogenic emissions from easterly and significant input from westerly (Li et al., 2013). On the whole, the BC concentrations over the TP varies with atmospheric circulation and upwind emission sources, and the high level of BC at QOMS suggest the significant influence of anthropogenic emissions from South Asia.

**3.3 Potential sources and transport mechanism of BC in different seasons**

The seasonal variation of BC concentrations was correlated with combustion intensity of sources and atmospheric

circulation. The "Atmospheric Brown Clouds" over South Asia contains large amounts of aerosol components such as the high loading emissions of BC from biomass burning, which can reach the TP within a few days (Ramanathan et al., 2005; Luthi et al., 2015; Ramanathan and Ramana, 2005). Previous study has identified biomass burning source contributing to BC aerosols in the Himalayas, and showed that the major fires were concentrated in March to June, besides, most vegetation fires occurred in the low elevation areas in South Asia and were mainly from croplands and forests (Vadrevu et al., 2012). Therefore, we further checked the biomass burning emissions in the Mt. Everest region and its vicinities using the active fire product from the MODIS data at four seasons (August 2015 to April 2016) provided by the FIRMS (Fig. 7). It is clearly understood there were large numbers of active fire spots in northern and central India, also in Pakistan and Nepal in winter and the pre-monsoon season, indicating that the agricultural combustion and forest fires contributed much to BC aerosols. During the monsoon season, no active fire spots distributed in South Asian region, representing the low biomass burning in that period.

To further explore the sources and the long-range transport mechanism of BC aerosols at QOMS, we calculated the frequency plots for 5-day backward trajectories arriving 1 km above the ground level (Fig. 8). During the non-monsoon seasons, air masses were affected by the westerly winds. The air masses reaching the Mt. Everest region were mostly from the northwest, indicating that the biomass burning emissions in Pakistan, northern Indian and Nepal could be transported to the Mt. Everest region. But for the difference of combustion intensity, the high concentrations of BC were found only during the pre-monsoon season. During the monsoon season, the southerly winds dominated in the Mt. Everest region and the air masses were mainly from the Arabian Sea and the Bay of Bengal with lots of moisture. At this period, the precipitation in the southern side of the Himalayas was above 1200 mm (Xu et al., 2014), which can improve the wet removal efficiency of BC. Moreover, the biomass combustion emissions in South Asia during this period were very low. Therefore, BC concentrations at QOMS were close to the background level during the monsoon season. Meanwhile, the local meteorological conditions also play a very important role in the transport of pollutants across the Himalayas from South Asia. Previous studies have shown that the local wind system was mainly composed by the uplift surface heating wind in the southern slope and downward glacier wind in the northern slope, which facilitates the exchange of air between bottom and upside of the atmosphere, also facilitates the coupling of airflow between the southern and northern slopes, which allows the pollutants from South Asia cross the Himalayas and transport to the TP from valley easily (Cong et al., 2015b; Zou et al., 2008; Tripathee et al., 2017; Chen et al., 2012; Dhungel et al., 2018).

**3.4 Pollution episodes analysis by WRF-Chem modeling**

In this section, we analyzed four pollution events with BC concentrations above 1000 ng/m$^3$ in detail, including event A during 8-10 June 2015, event B during 19-22 March 2016, event C during 9-18 April 2016, and event D during 11-14 April 2017. Fig. 9 shows the spatial characteristic of the WRF-Chem modeled surface BC concentrations during the four pollution episodes. It can be seen that, the high values of surface BC concentrations always appeared in South Asia, although the highvalue centers changed in different pollution events. For event A, the most serious pollution appeared in Nepal and northern India. Relatively, there were less BC nearby the Mt. Everest in event B. However, for event C, the high-value areas for BC concentrations were mainly along the southern slope of the Himalayas in Nepal and in the east of India, which can result in a great impact for BC concentrations in the Mt. Everest region. In event D, the high BC occurred in Nepal and some parts of India.

The sources and transport mechanism of BC aerosols during these pollution episodes can be indicated by analyzing the air flow. Fig. 10 shows the variation of BC concentrations and wind field at different altitudes in the atmosphere (850 hPa, 500 hPa, 200 hPa). For event A during the monsoon season, there was a cyclone in northern India at 850 hPa, which moved near-surface BC aerosols upward and then transported to the Mt. Everest region by the southward winds at 500 hPa and 200 hPa. For events B-D in the pre-monsoon season, northwesterly winds prevailed in South Asia at 850 hPa and brought BC from northern India to the southern slope of the Himalayas, and westerly winds at 500 hPa and 200 hPa can transport relatively less BC from northwestern India and Central Asia to the Mt. Everest region. Previous studies also pointed out that BC can be transported across the Himalayas to the Mt. Everest region by the mountain-valley wind system (Zou et al., 2008; Cong et al., 2015b; Dhungel et al., 2018). Thus, we needed to further analyze the impact of the mountain-valley wind on the transportation of BC. Fig. 11 shows the vertical profile of BC concentration among the QOMS's longitude of 86.95°E. During event A, high concentrations of BC appeared in the upper atmosphere of South Asia and many BC aerosols were transported to most parts of the TP (Fig. 11a), due to the large-scale transport process. However, for events B-D, high concentrations of BC occurred along the southern slope of the Himalayas and BC aerosols were only transported to a few areas of the northern slope of the Himalayas such as the Mt. Everest region (Fig. 11b-d), caused by the local mountain-valley wind. As shown in Fig. S2, for events B-D, the mountain wind in the southern side of the Himalayas can move BC aerosols up in the daytime and the down valley wind can make it fall down in the Mt. Everest region at night.

To sum up, we found that the transport processes of BC aerosols from South Asia to the QOMS were different as seasons varying. In the monsoon season such as event A, BC aerosols were moved upward by the cyclone in the lower atmosphere and were transported to QOMS by the southward winds in the upper atmosphere. However, in the pre-monsoon season such as events B-D, the mountain-valley wind played an import role in the BC aerosols transported from the southern slope of the Himalayas to the Mt. Everest region.

**4. Conclusions**

In this study, BC concentrations were measured from 15 May 2015 to 31 May 2017 at QOMS in the south edge of the TP, and monthly, daily, and diurnal variation of BC concentrations were calculated to investigate the temporal characteristics and potential sources of BC at QOMS. The results showed that the monthly mean BC concentrations reached the highest value in the pre-monsoon season (923.1 ± 685.8 ng/m$^3$ in April) and the lowest value in the monsoon season (88.5 ± 29.8 ng/m$^3$).

Average daily BC concentration was equal to 298.8 ± 341.3 ng/m$^3$, with a maximum in the pre-monsoon season (2772.3 ng/m$^3$) and a minimum in the monsoon season (36.4 ng/m$^3$). The diurnal variation of BC concentrations in the pre-monsoon season showed significant high values from mid-night to noon, and there was an increasing trend in the afternoon during the non-monsoon periods, implying the potential origin of BC are from the long-range transport. BC concentrations appeared two peaks in the morning and after the noon during the monsoon period, might be affected by the local anthropogenic activities.

The seasonal cycle of BC concentrations at QOMS was closely correlated with the variation of atmospheric circulation and combustion emissions in South Asia. In the non-monsoon seasons, affected by westerly, the air masses in the Mt. Everest region were largely from Pakistan, northern Indian, and Nepal, where existed high loading emissions of vegetation fires. In the monsoon season, the southerly winds were dominated in the Mt. Everest region and the air masses were mainly from the Arabian Sea and the Bay of Bengal. Under intense precipitation scavenging of BC and extremely low level of the combustion emissions in South Asia, BC concentrations at QOMS were close to the background level in the monsoon season.

For four heavy pollution episodes occurred at QOMS with BC concentrations above 1000 ng/m$^3$, we found that the transport processes of BC aerosols from South Asia to the Mt. Everest region were different as seasons varying. In the monsoon season (take the pollution event during 8-10 June 2015 as an example), BC aerosols were efficiently driven upward by the cyclone in the lower atmosphere in South Asia and transported to the Mt. Everest region by the southward winds in the upper atmosphere. However, during the pre-monsoon season (take the other three pollution events as example), the mountain-valley wind played an import role in the BC aerosols cross the Himalayas and were transported 
[revised manuscript text omitted]

Xu, B.-Q., Wang, M., Joswiak, D. R., Cao, J.-J., Yao, T.-D., Wu, G.-J., Yang, W., and Zhao, H.-B.: Deposition of anthropogenic aerosols in a southeastern Tibetan glacier, J. Geophys. Res., 114, D17209, http://doi.org/10.1029/2008JD011510, 2009.

Xu, C., Ma, Y. M., Panday, A., Cong, Z. Y., Yang, K., Zhu, Z. K., Wang, J. M., Amatya, P. M., and Zhao, L.: Similarities and differences of aerosol optical properties between southern and northern sides of the Himalayas, Atmos. Chem. Phys., 14, 3133-3149, http://doi.org/10.5194/acp-14-3133-2014, 2014.

Xu, Y., Ramanathan, V., and Washington, W. M.: Observed high-altitude warming and snow cover retreat over Tibet and the Himalayas enhanced by black carbon aerosols, Atmos. Chem. Phys., 16, 1303-1315, http://doi.org/10.5194/acp-16-1303-2016, 2016.

Yang, J., Duan, K., Kang, S., Shi, P., and Ji, Z.: Potential feedback between aerosols and meteorological conditions in a heavy pollution event over the Tibetan Plateau and Indo-Gangetic Plain, Clim. Dyn., 48, 2901-2917, http://doi.org/10.1007/s00382-016-3240-2, 2017.

Yang, J., Kang, S., Ji, Z., and Chen, D.: Modeling the Origin of Anthropogenic Black Carbon and Its Climatic Effect Over the Tibetan Plateau and Surrounding Regions, J. Geophys. Res.: Atmos., n/a-n/a, http://doi.org/10.1002/2017JD027282, 2018.

Yang, M., Howell, S. G., Zhuang, J., and Huebert, B. J.: Attribution of aerosol light absorption to black carbon, brown carbon, and dust in China - interpretations of atmospheric measurements during EAST-AIRE, Atmos. Chem. Phys., 9, 2035-2050, http://doi.org/10.5194/acp-9-2035-2009, 2009.

Yang, S., Xu, B., Cao, J., Zender, C. S., and Wang, M.: Climate effect of black carbon aerosol in a Tibetan Plateau glacier, Atmos. Environ., 111, 71-78, http://doi.org/10.1016/j.atmosenv.2015.03.016, 2015.

Yao, T., Thompson, L. G., Mosbrugger, V., Zhang, F., Ma, Y., Luo, T., Xu, B., Yang, X., Joswiak, D. R., Wang, W., Joswiak, M. E., Devkota, L. P., Tayal, S., Jilani, R., and Fayziev, R.: Third Pole Environment (TPE), Environ. Dev., 3, 52-64, http://doi.org/10.1016/j.envdev.2012.04.002, 2012.

Yasunari, T. J., Bonasoni, P., Laj, P., Fujita, K., Vuillermoz, E., Marinoni, A., Cristofanelli, P., Duchi, R., Tartari, G., and Lau, K. M.: Estimated impact of black carbon deposition during pre-monsoon season from Nepal Climate Observatory – Pyramid data and snow albedo changes over Himalayan glaciers, Atmos. Chem. Phys., 10, 6603-6615, http://doi.org/10.5194/acp-10-6603-2010, 2010.

Zhang, Q., Streets, D. G., Carmichael, G. R., He, K. B., Huo, H., Kannari, A., Klimont, Z., Park, I. S., Reddy, S., Fu, J. S., Chen, D., Duan, L., Lei, Y., Wang, L. T., and Yao, Z. L.: Asian emissions in 2006 for the NASA INTEX-B mission, Atmos. Chem. Phys., 9, 5131-5153, http://doi.org/10.5194/acp-9-5131-2009, 2009.

Zhang, R., Wang, H., Qian, Y., Rasch, P. J., Easter, R. C., Ma, P. L., Singh, B., Huang, J., and Fu, Q.: Quantifying sources, transport, deposition, and radiative forcing of black carbon over the Himalayas and Tibetan Plateau, Atmos. Chem. Phys., 15, 6205-6223, http://doi.org/10.5194/acp-15-6205-2015, 2015.

Zhang, X., Ming, J., Li, Z., Wang, F., and Zhang, G.: The online measured black carbon aerosol and source orientations in the Nam Co region, Tibet, Environ. Sci. Pollut. Res., 24, 25021-25033, http://doi.org/10.1007/s11356-017-0165-1, 2017a.

Zhang, Y., Kang, S., Cong, Z., Schmale, J., Sprenger, M., Li, C., Yang, W., Gao, T., Sillanpää, M., Li, X., Liu, Y., Chen, P., and Zhang, X.: Light-absorbing impurities enhance glacier albedo reduction in the southeastern Tibetan plateau, J. Geophys. Res.: Atmos., 122, 6915-6933, http://doi.org/10.1002/2016JD026397, 2017b.

Zhao, S., Ming, J., Xiao, C., Sun, W., and Qin, X.: A preliminary study on measurements of black carbon in the atmosphere of northwest Qilian Shan, J. Environ. Sci., 24, 152-159, http://doi.org/10.1016/s1001-0742(11)60739-0, 2012.

Zou, H., Zhou, L., Ma, S., Li, P., Wang, W., Li, A., Jia, J., and Gao, D.: Local wind system in the Rongbuk Valley on the northern slope of Mt. Everest, Geophys. Res. Lett., 35, L13813, http://doi.org/10.1029/2008gl033466, 2008.

**Table 1. Mean BC concentrations at QOMS and compared with other remote sites.**

| Name | Location | Sample | Sampling period | BC or EC (ng/m$^3$) | Reference |
|---|---|---|---|---|---|
| QOMS | Southern TP (28.36°N, 86.95°E, 4276m) | AE33 | May 2015-Apr 2016 | 298.8 ± 341.3 | This paper |
| QOMS | Southern TP (28.36°N, 86.95°E, 4276m) | TSP | Aug 2009-Jul 2010 | 250 ± 220 | Cong et al. (2015) |
| Ranwu | Southeast TP (29.32°N, 96.96°E, 4600m) | AE31 | Nov 2012-Jun 2013 | 139.1 | Wang et al. (2016) |
| Lhasa | Southwest TP (29.65°N, 91.03°E, 3642m) | PM 10 | May 2013-Mar 2014 | 2310 | Li et al. (2016) |
| Nam Co | Central TP (30.46°N, 90.59°E, 4730m) | TSP | Jan-Dec 2012 | 190 | Wan et al. (2015) |
| Qilianshan | Northern TP (39.50°N, 96.51°E, 4214m) | AE31 | May 2009-Mar 2011 | 48 | Zhao et al. (2012) |
| Qinghai Lake | Northeast TP (37.00°N, 99.90°E, 3200m) | PM 2.5 | Jul-Aug 2010 | 370 | Li et al. (2013) |
| Muztagh Ata | Northwest TP (38.29°N, 75.02°E, 4500m) | TSP | Dec 2003-Feb 2006 | 55 | Cao et al. (2009) |
| NCO-P, Nepal | Southern Himalayas (27.95°N, 86.82°E, 5079m) | PM 1 | Mar 2006-Feb 2008 | 160.5 ±296.1 | Marinoni et al. (2010) |
| Manora Peak, India | Central Himalayas (29.40° N, 79.50° E, 1950m) | TSP | Feb 2005–Jul 2008 | In the range of 140-7600 | Ram et al. (2010) |

[Figure]

**Figure 1: Location of the sampling site.**

[Figure]

**Figure 2: Variations of temperature, humidity, wind speed, wind direction, and precipitation at QOMS from May 2015 to May 2017.**

[Figure]

**Figure 3: (a) Monthly mean BC concentrations at QOMS from May 2015 to May 2017 in this study; (b) Monthly mean EC at NCO-P from March 2006 to February 2008 from Marinoni et al. (2010); (c) Monthly mean EC at QOMS from August 2009 to July 2010 from Cong et al. (2015); (d) Monthly mean EC at Nam Co station from January to December 2012 from Wan et al. (2015).**

[Figure]

505

506 **Figure 4: Daily mean BC concentrations at QOMS during study period (the gray bars represent the continuous high values more**
507 **than 1000 ng/m³).**

508

[Figure]

**Figure 5: Diurnal variation of BC concentrations (every half an hour) at QOMS during study period.**

[Figure]

512

513 **Figure 6. Wind direction frequency at QOMS in the pre-monsoon season (a) 0:00-12:00; (b) 12:00-24:00.**

514

[Figure]

**Figure 7. The distribution of fire spots in different seasons from August 2015 to April 2016.**

[Figure]

**Figure 8. Frequency plots for 5-day back trajectories calculated by HYSPLIT model at QOMS in different seasons from August 2015 to April 2016.**

[Figure]

**Figure 9. Mean BC concentration simulated by WRF-Chem model at QOMS and its vicinities: (a) event A; (b) event B; (c) event C; (d) event D.**

[Figure]

Figure 10. Mean BC concentration and wind at 850 hpa, 500 hpa, and 200 hpa simulated by WRF-Chem model at QOMS and its vicinities: event A (the first row); event B (the second row); event C (the third row); event D (the last row).

[Figure]

**Figure 11. Vertical profiles of mean BC concentration among the QOMS's longitude of 86.95°E: (a) event A; (b) event B; (c) event C; (d) event D.**

---

## Author Comment (AC1) · 13 Jun 2018

We greatly appreciate the reviewers' valuable and constructive suggestions concerning our manuscript (ID: acp-2018-183). The point-by-point reply to the comments are as follow:

Response to Referee's Comments 1

The manuscript investigated the seasonal and diurnal variations of BC and its potential source regions in the Tibetan Plateau. It is significant to research the effect of pollutant aerosol on the Tibetan Plateau. But there are some comments to improve the

manuscript. 1. Merge Table 1 in Figure 1. So the reader can directly see the distribution of BC in the Tibetan Plateau. Author response: Thanks for the reviewer's advice. We have merged Table 1 in Figure 1 and supplemented sites information in Table S1.

2. Figure 2 and Figure 6 are redundant. The weather data is not significant in the research. Author response: We have put Figure 2 in the supplement materials. Because the diurnal variation in BC in the pre-monsoon season showed obvious high values, we presented the wind direction frequency at QOMS in the pre-monsoon season (Figure 6, but in the new revision it was changed to Figure 5) to help us better understand the sources of BC during this period.

3. Add some important references to compare with the results, e.g. Xin et al., BAMS, 2015. Author response: According to the reviewer's suggestion, we have now added some important references (e.g., Xin et al., 2015) and compared them with our results in the revised manuscript. Please see Lines 228-229 and 235-237.

4. Compare the BC concentration of WRF-Chem with the observation in the Tibetan Plateau. Author response: Thanks for the reviewer's advice. We have compared the simulated BC concentrations with the observations at QOMS during the four heavy pollution episodes. The WRF-Chem model could capture the variation trends of BC concentrations at this sampling site, with correlation coefficients all above 0.8 for these four pollution episodes as shown in Figure S3. The relevant statement has been added in Lines 284-287.

Please also note the supplement to this comment:
https://www.atmos-chem-phys-discuss.net/acp-2018-183/acp-2018-183-AC1-supplement.pdf

———————————————

[Figure]

**Fig. 1.** Distribution of BC concentrations over the TP based on the observed values at QOMS in this study (black circle) and from previous studies (red circles), i.e., at QOMS (Cong et al., 2015a), Nam Co (Wan

**Fig. 2.** Comparisons between simulated BC concentrations and the observation at QOMS during the four pollution episodes: (a) event A, (b) event B, (c) event C, and (d) event D.

---

## Author Comment (AC2) · 13 Jun 2018

We greatly appreciate the reviewers' valuable and constructive suggestions concerning our manuscript (ID: acp-2018-183). The point-by-point reply to the comments are as follow:

Response to Referee's Comments 2

The authors conducted a detailed analysis of the BC concentration measurements at the Qomolangma station. The measurement, with a high temporal resolution and a relatively long period, provides very valuable information for the understanding of BC

sources and transport to the Himalayas. The authors further combined observations with model simulations to investigate the BC transport mechanism. The analysis is comprehensive and the manuscript is generally well written. Before it can be considered for publication, I have a few comments and suggestions.

1. Section 2.2: Since different measurement methods may lead to quite different BC or EC concentrations, I suggest adding some discussions in this section on the possible difference in measured BC concentrations between the AE-33 used in this study and some widely-used methods from previous studies (e.g., thermal-optics method, SP2, etc.).

Author response: Thanks for reviewer's suggestion. We have added some discussions about the three commonly used methods as follow: There are several available methods capable of measuring BC concentrations, and these methods can be classified into three categories. First is the thermal/optical method, which uses a quartz filter to collect aerosols, and they are thermally volatilized in several temperature steps (Schauer et al., 2003). The signals of evolving carbon measured by thermal/optical transmission (TOT) or thermal/optical reflectance (TOR) can be converted to the concentration of BC (Chow et al., 1993; Chow et al., 2001). However, the time difference between sampling and detection, the impact of mineral dust, and the determination of the split between organic carbon (OC) and elemental carbon (EC, the same as BC) can cause deviations (Li et al., 2017a; Schauer et al., 2003). The second category is the technique of the single particle soot photometer (SP2), which can quantify BC by laser-induced incandescence because BC is the predominant refractory absorbing aerosol, which can be heated by an intense laser beam and emit significant thermal radiation (Stephens et al., 2003). This method measures the mass of BC in individual particles, but the accuracy depends on the selected calibration material (Schwarz et al., 2010; Laborde et al., 2012). Finally, the optical method measures the reduction in light intensity induced by BC aerosols collected on the sampling medium (Hansen et al., 1984; Petzold and Schonlinner, 2004). The Aethalometer is a widely used instrument based on the optical method that can provide real-time BC concentration measurements, but all filter-based optical methods exhibit loading effects that can lead to the underestimation of BC concentrations (Bond et al., 1999; Virkkula et al., 2007; Park et al., 2010; Hyvarinen et al., 2013; Drinovec et al., 2015). However, the newly developed Aethalometer model AE-33 uses a real-time loading effect compensation algorithm that can provide high-quality data, which is very helpful for the accurate determination of BC concentrations and source apportionment (Drinovec et al., 2015) (Lines 95-111).

2. Section 3.4: The authors did a detailed analysis on possible BC sources and transport mechanisms for four pollution events, which is great. However, the evaluation of WRF-Chem model simulation seems missing here. Without knowing the model performance, it is difficult to be convinced by the source and transport analysis of model results. At least, the authors could compare modeled BC concentrations at this site with their observations. If possible, the modeled wind and precipitation can be also evaluated against some reanalysis or satellite products. If this takes too much time, the authors could also cite and discuss some previous studies where the WRF-Chem simulations have been evaluated in the TP and surrounding regions.

Author response: According to the reviewer's advice, we compared the simulated BC concentrations with the observations at QOMS during the four heavy pollution episodes, please find in Lines 284-287 and Figure S3. The WRF-Chem model could capture the variation trends of BC concentrations at this sampling site, with correlation coefficients all above 0.8 for the four pollution episodes. We also compared the WRF-Chem simulated 500 hPa wind and 500 hPa relative humidity with the ERA-Interim reanalysis data during the non-monsoon season. As shown in Figure S4, the simulated results had a good agreement with the reanalysis data in spatial distribution. Additionally, compared with the observations from 73 national meteorological stations, the WRF-Chem simulation results represented well the monthly variation of precipitation (Figure S5). Moreover, the simulation setup and selection of parameterization schemes in this study were according to Yang et al. (2018)'s study, which pointed out that the WRF-Chem model can capture key spatiotemporal variations of wind and precipitation over the TP and its adjacent regions, compared with independent observations and reanalysis data. We have added these comparisons in Lines 287-289.

3. Line 15: "... concentrations were significantly greater from mid-night to noon. . ." This sentence is a little confusing. Do you mean "concentrations increased from mid-night to noon"?

Author response: The meaning of this sentence is that the BC concentrations remained significantly high from midnight to noon in the pre-monsoon season, compared with other times of a day. We have corrected this sentence in Lines 14-15.

4. Line 16: ". . ., implying the potential contribution from the long-range transport." It is not very straightforward for readers to understand why such diurnal variation implies the contribution from the long-range transport. Could you please rephrase the sentence and clarify the point?

Author response: The BC concentrations remained significantly high from midnight to noon in the pre-monsoon season. Meanwhile, the westerly winds prevailing during this period provided the potential possibility for pollutants to be transported across the Himalayas from long-distance sources to QOMS along the valley. We have rephrased this sentence in Lines 14-17.

5. Line 40: For the authors' information, a recent study (Lee et al., 2017) investigated BC deposition effects on reducing snow albedo over the Tibetan plateau based on satellite observation analysis. This study can be cited here.

Author response: Lee et al. (2017) revealed the impact of absorbing aerosol deposition on snow albedo reduction over the TP, which can support our statement that BC is an important contributor to rapid shrinking of glaciers over the TP and we have cited this study in Line 42.

6. Line 57: For the authors' information, a recent study (He et al., 2014b) has also used a global CTM to investigate the sources of BC over the Tibetan Plateau based on a tag-tracer technique, which can be cited here.

Author response: Considering the reviewer's suggestion, we have cited the study of He et al. (2014b) in Lines 59-60 to support our statement.

7. Lines 102–112: It would be more informative if the authors could provide the uncertainty/accuracy associated with this algorithm for BC concentration calculation.

Author response: Previous studies demonstrated that more accurate BC concentration could be obtained by the new real-time compensation algorithm of AE-33, which is based on the dual-spot technology and allows extrapolation to zero loading (Drinovec et al., 2015; Crenn et al., 2015; Zhu et al., 2017). Furthermore, the comparison between AE-33 and earlier Aethalometer models and other filter-based absorption photometers showed the well performance of this new algorithm (Drinovec et al., 2015; Rajesh and Ramachandran, 2018). We have added these discussions in Lines 132-137.

8. Section 2.3 "Model simulation": A number of studies (e.g., Flanner et al., 2007; Liou et al., 2014; He et al., 2017) have shown significant effects from BC in snow on albedo simulations. This albedo effect and feedback may exert an important impact on model simulations. Did the authors include such "dirty snow" effect in the WRF-Chem simulations? I suggest adding a brief discussion on this issue. .

Author response: Thank for the reviewer's inspiring suggestion. Previous studies (e.g., Flanner et al., 2007; Liou et al., 2014; He et al., 2017) have shown significant effects from BC in snow on albedo simulations, and this albedo effect and feedback may exert an important impact on model simulations. But the WRF-Chem model used in this study cannot simulate the radiative effect of absorbing aerosols, because the SNICAR (snow, ice, and aerosol radiative) model is not fully coupled into the WRF-Chem. In the future, we will try our best to connect the WRF-Chem atmospheric aerosol deposition with the SNICAR model to analyze the radiative effect of BC in the snow. Additionally, we compared our results with reanalysis data and in-situ observations (Figure S3,

Figure S4, and Figure S5). The comparison suggested that the WRF-Chem can capture the key spatiotemporal characteristics of meteorological elements and surface BC concentrations in this study area.

9. Line 183: "..., which might be owing to the surrounding local emissions." Is there any reference/observation showing surrounding emissions? Is there any populated city or town around the observational site? More information would better convince readers.

Author response: There are several villages located north (approximately 5 km away) of the observational site (QOMS), and the uplifted valley wind from the north in the morning could bring the short-distance emissions from local cooking or heating to QOMS. We have added this information in Lines 210-211.

Please also note the supplement to this comment:
https://www.atmos-chem-phys-discuss.net/acp-2018-183/acp-2018-183-AC2-supplement.pdf

**Fig. 1.** Comparisons between simulated BC concentrations and the observation at QOMS during the four pollution episodes: (a) event A, (b) event B, (c) event C, and (d) event D.

**Fig. 2.** Mean wind (m s-1) and relative humidity (RH, %) at 500 hPa during the non-monsoon season from the WRF-Chem simulation (a, c) and the ERA-Interim (b, d), respectively.

[Figure]

Monthly precipitation

Precipitation (mm)

OBS
WRF-Chem

**Fig. 3.** Monthly mean precipitation in 2013, averaged at 73 sites over the TP. Data are from the observations at national stations (OBS) and the model simulation in this study (WRF-Chem).

**Supplement:**

[revised manuscript text omitted]

---

## Author Comment (AC3) · 13 Jun 2018

We greatly appreciate the reviewers' valuable and constructive suggestions concerning our manuscript (ID: acp-2018-183). The point-by-point reply to the comments are as follow:

Response to Referee's Comments #3

This paper is talking about mass concentrations, temporal variations and source regions along with transported mechanisms of airborne BC observed in Mountain Everest region. The authors have revised this paper followed by the reviewer's comments. I

suggest that this paper can be published to ACP and some minor comments are listed as:

1. Line 16, change "...in the non-monsoon season" to "...during the monsoon season".

Author response: In the new revision, we have rewritten this sentence to make the explanation more clearly in Lines 14-17.

2. Line 78-79. What is the time resolution for measurements in meteorological parameters? The authors should state.

Author response: The meteorological parameters at QOMS were measured by an automatic weather station at QOMS with 10 min time intervals. We have added this statement in Line 81.

3. Line 129-133. The authors should cite papers for utilizations of HYSPLIT model.

Author response: Stein et al. (2015) presented the HYSPLIT model developments and applications, and we have cited it in Lines 154-155. Moreover, Xu et al. (2014) used the HYSPLIT model to calculate the backward trajectories of the air masses at QOMS. Our selection of parameters in the HYSPLIT model was according to Xu et al. (2014)'s study, and we have cited it in Lines 157-158.

4. Line 141. Please replace "......at QOMS (Fig. 3c) (Cong et al., 2015a)" by "......at QOMS from August 2009 to July 2010 (Fig. 3c) (Cong et al., 2015a)".

Author response: We have replaced the sentence "...at QOMS (Fig. 3c) (Cong et al., 2015a)" by "......at QOMS from August 2009 to July 2010 (Fig. 2c) (Cong et al., 2015a)" in Line 169. And we have also added the research period of other two comparison sites, including the EC at NCO-P station from March 2006 to February 2008 and the EC at Nam Co station from January to December during 2012 in Line 168 and 174, respectively.

5. Line 161. I think the decreased BC during the monsoon is due to washout by

precipitation. So, please remove humidity in the sentence of "...increasing humidity and..."

Author response: We have removed humidity in the sentence of "...increasing humidity and precipitation ..." in Line 190.

6. Fig. 11. It is better to point out the location of QOMS in all panels of this figure for the reader to follow it.

Author response: Thanks for the reviewer's advice. We have marked the location of QOMS in all panels of Figure 10. Because we have moved a figure into supplement materials, the Figure 11 in the original manuscript was changed to Figure 10 in the new revision.

Please also note the supplement to this comment:
https://www.atmos-chem-phys-discuss.net/acp-2018-183/acp-2018-183-AC3-supplement.pdf

[Figure]

**Fig. 1.** Vertical profiles of mean BC concentration among the QOMS's longitude of 86.95°E: (a) event A; (b) event B; (c) event C; (d) event D.

---

## Author Comment (AC4) · 13 Jun 2018

We greatly appreciate the reviewers' valuable and constructive suggestions concerning our manuscript (ID: acp-2018-183). The point-by-point reply to the comments are as follow:

Response to Referee's Comments 4

1. This manuscript present analysis of the high-resolution measurement of black carbon (BC) at Qomolangma (Everest) station of Chinese Academy of Sciences during 15 May 2015 to 31 May 2017, together with model simulations to investigate the possible

transport mechanisms of BC. Generally, the manuscript is well organized, but many sentences and even paragraphs still need to be clarified or improved. Though I have marked many places in the text, I believe that there are still more problematic phases or sentences to be identified and corrected. I suggest that the whole text should be carefully checked and improved with the help of an English editor.

Author response: Thanks for reviewer's advices, we have very carefully checked the whole text and corrected the problematic phrases or sentences and clarified the explanations marked in the text. Moreover, this manuscript has been edited for proper English language, grammar, punctuation, spelling, and overall style by the highly qualified native English speaking editor at American Journal Experts. We have uploaded the editorial certificate file in the attachment.

2. Some of the explanations are not convincing. For example, in line 182 of page 6, it reads "The valley wind from north in the morning, could bring the short-distance emissions from local cooking or heating to QOMS. BC concentrations appeared two peaks in the morning and after the noon in the monsoon season, which might be owing to the surrounding local emission." Why it occurred in the morning and afternoon in the monsoon season, not other times and in other seasons? It should be clarified to what extent the daily and seasonal values and patterns obtained in this study are influenced by local emissions.

Author response: Thanks for reviewer's kind suggestion. We have checked the Section 3.1.3 (Diurnal variation in BC) and rewrite some explanations in Lines 200-204, 210-213, 222-226, and 317-322. In the pre-monsoon period, the BC concentrations remained significantly high from midnight to noon and increased gradually after the lowest value at approximately 15:00. Elevated BC concentrations were also observed in the afternoon during the post-monsoon and winter seasons. According to previous studies, the significantly increased BC was closely linked with the strengthened down-valley wind in the afternoon and at night (such as in the pre-monsoon season), which could deliver the trans-Himalayan pollutions to QOMS. The high values of diurnal BC

concentrations from midnight to noon at QOMS were related to down-valley wind transport as well as stable atmosphere in the pre-monsoon season. However, during the monsoon period, the BC concentrations were significantly lower than those during the other seasons but peaked in the morning and in the afternoon, which might due to the local cooking emissions carried by the valley wind from the north. There are several villages located north (approximately 5 km away) of QOMS.

3. Section 3.2 is not well written. What do you want to say through these comparisons?

Author response: Considering the reviewer's suggestions, we have rewritten Section 3.2 in Lines 228-245. In Section 3.2, we hope to better understand the BC loading level and investigate its potential emission sources at QOMS by comparing our results with previous studies at other sites over the TP. The rewritten Section 3.2 is as follow: A previous study have revealed that low BC concentrations in China can be found on the TP, with values of approximately 200-1000 ng/m3 in PM2.1 and 300-1500 ng/m3 in PM9.0 (Xin et al., 2015). To better understand the BC loading level, we compared our results with previous studies from other locations over the TP. As shown in Fig. 1, the BC concentrations at Muztagh Ata and Qilian Shan presented low values, which can be regarded as the background concentration level for inland Asia (Cao et al., 2009; Zhao et al., 2012). In contrast, the BC concentration at Lhasa city was higher than that at other sites on the TP, which was mainly contributed by local fossil fuel combustion (Li et al., 2016b). In addition, under the impact of the long-range transport of anthropogenic emissions from the east and significant dust input from the west, the BC concentration at Qinghai Lake also showed a relatively high value (Li et al., 2013). The BC concentration at Beiluhe was slightly higher than that at Qinghai Lake, mainly from the arid regions in northwestern China in spring and from the southern slope of the Himalayas in winter (Wang et al., 2016). Therefore, the long-range transportation from Central Asia and East Asia contributed greatly to the BC aerosols in the northern TP. For the sites in the central and southeastern regions on the TP (e.g., Nam Co and Ranwu), which are isolated from anthropogenic activities with relatively clean atmospheric environments, the BC concentrations at these two sites were above 130 ng/m3, likely due to the influence of long-range transport from South Asia (Wan et al., 2015; Wang et al., 2016). Compared with the locations on the southern slope of the Himalayas (e.g., NCO-P and Manora Peak), the BC concentration at QOMS was close to that at NCO-P but much lower than that at Manora Peak, which is near the polluted areas in South Asia and largely affected by anthropogenic emissions (Marinoni et al., 2010; Ram et al., 2010). This implies that the combustion emissions from South Asia affect not only the lower latitudes in the vicinity but also the higher latitudes in the Himalayas and the interior of the TP due to long-range transport.

4. The authors should indicate what is new in this study. It seems to me that most of the results are similar to those obtained in previous studies, although different instruments and models might be used in different studies.

Author response: Previous studies of BC in this region were mainly based on the thermal/optical method with a lower time resolution, through the quartz filter sampling. And the detailed investigation about diurnal and seasonal variations in BC still lacks in this region. In this study, we have presented the real-time data of BC concentrations from 15 May 2015 to 31 May 2017, which can provide more information in diurnal and seasonal variations as well as pollution events, and help us improve the knowledge about sources and transport mechanisms. In addition to the analysis of fire spots and backward trajectories based on the previous studies, we have also used WRF-Chem model and simulated the specific transport processes during the pollution episodes in different seasons, including the horizontal and vertical transport. That is helpful to clarify the transport mechanisms of trans-Himalayan BC aerosols from South Asia.

5. More corrections and comments are marked in the text.

Author response: Thanks for reviewer's patient and valuable revision. We have corrected the problematic phrases or sentences and clarified the explanations marked in the text. The revised manuscript is uploaded, please find in the supplementary files.

Please also note the supplement to this comment:
https://www.atmos-chem-phys-discuss.net/acp-2018-183/acp-2018-183-AC4-
supplement.zip
* * *

---

## Referee Report (RR1)

All my questions and suggestions have been answered or clarified in previous revision. The quality of the manuscript has been improved significantly. I have two more comments:

1. A lot of terms, such as "noon', "midnight', etc., are used to refer local times, but it seems to me (at least from Figure 4) that all these terms are defined with Beijing times. Because there are about three hours difference between the local time and Beijing Time, the descriptions in the text should be modified correspondingly. For example, the "noon" in Beijing Time is actually the local morning at QOMS.

2. In lines 13-14 in the abstract: "Monthly and daily mean BC concentrations reached the highest values in the pre-monsoon season, which were at least one magnitude higher than the lowest values in the monsoon season": Firstly, it is inadequate to compare the mean values in the pre-monsoon with the lowest values in the monsoon season; secondly, it should be "one order of magnitude higher than …".

---

## Author Response (AR2)

Dear editor and reviewers,

We are particularly grateful for your efforts and the time you have invested in the review processes. We also very appreciate the reviewers' valuable and constructive suggestions concerning our manuscript "Concentration, temporal variation and sources of black carbon in the Mount Everest region retrieved by real-time observation and simulation" (ID: acp-2018-183). Our changes were marked in yellow background in the revised manuscript. We hope that you will find our manuscript significantly improved as a result of these revisions. Our point-by-point reply to the comments are as follow:

**Response to Referee's Comments #4**

All my questions and suggestions have been answered or clarified in previous revision. The quality of the manuscript has been improved significantly. I have two more comments:

1. A lot of terms, such as "noon', "midnight', etc., are used to refer local times, but it seems to me (at least from Figure 4) that all these terms are defined with Beijing times. Because there are about three hours difference between the local time and Beijing Time, the descriptions in the text should be modified correspondingly. For example, the "noon" in Beijing Time is actually the local morning at QOMS.

**Author response:** Thanks for the reviewer's advice. We have revised the descriptions of the terms (such as "noon", "midnight", etc.,), which were defined with Beijing Time in the previous manuscript. Beijing Time is two hours earlier than local time, and we have modified the descriptions in Section 3.1.3 in this revision. For example, "from midnight to noon" has been changed to "from late night to morning" (00:00-12:00 BJT) in Line 202 and "from noon to midnight" has been changed to "from late morning to night" (12:00-24:00 BJT) in Line 224. In addition, the modifications for these terms in the Abstract and Conclusions sections are shown in Line 16, 18, 321, and 322.

2. In lines 13-14 in the abstract: "Monthly and daily mean BC concentrations reached the highest values in the pre-monsoon season, which were at least one magnitude higher than the lowest values in the monsoon season": Firstly, it is inadequate to compare the mean values in the pre-monsoon with the lowest values in the monsoon season; secondly, it should be "one order of magnitude higher than …".

**Author response:** According to the reviewer's suggestion, we have clarified the explanations in Lines 13-15. And the rewritten sentences are as follow:

[revised manuscript text omitted]